# Learning Structure from the Ground up—Hierarchical Representation Learning by Chunking

**Shuchen Wu**
Computational Principles of Intelligence Lab
Max Planck Institute for Biological Cybernetics
Tübingen, Germany
shuchen.wu@tuebingen.mpg.de

**Noémi Éltető**
Department of Computational Neuroscience
Max Planck Institute for Biological Cybernetics
Tübingen, Germany
noemi.elteto@tuebingen.mpg.de

**Ishita Dasgupta**
Computational Cognitive Science Lab
Department of Psychology
Princeton University
dasgupta.ishita@gmail.com

**Eric Schulz**
Computational Principles of Intelligence Lab
Max Planck Institute for Biological Cybernetics
Tübingen, Germany
eric.schulz@tuebingen.mpg.de

## Abstract

From learning to play the piano to speaking a new language, reusing and recombining previously acquired representations enables us to master complex skills and easily adapt to new environments. Inspired by the Gestalt principle of *grouping by proximity* and theories of chunking in cognitive science, we propose a hierarchical chunking model (HCM). HCM learns representations from non-i.i.d. sequential data from the ground up by first discovering the minimal atomic sequential units as chunks. As learning progresses, a hierarchy of chunk representations is acquired by chunking previously learned representations into more complex representations guided by sequential dependence. We provide learning guarantees on an idealized version of HCM, and demonstrate that HCM learns meaningful and interpretable representations in a human-like fashion. Our model can be extended to learn visual, temporal, and visual-temporal chunks. The interpretability of the learned chunks can be used to assess transfer or interference when the environment changes. Finally, in an fMRI dataset, we demonstrate that HCM learns interpretable chunks of functional coactivation regions and hierarchical modular and sub-modular structures supported by the neuroscientific literature. Taken together, our results show how cognitive science in general and theories of chunking in particular can inform novel and more interpretable approaches to representation learning.

## 1 Introduction

Sequential data in our everyday life is often hierarchically structured. From streaming this sequential sensory perceptual data, we can identify repeated patterns – and bootstrap these to recognize higher order patterns. In cognitive science, identifying repeated, invariant patterns from sequences in units is known as *chunking*. To get an intuition for chunking, try to read through the following sequence

36th Conference on Neural Information Processing Systems (NeurIPS 2022).

of letters: "DFJKJKJKDFDFJKJKDFDF". Upon reaching the end, if you were asked to repeat the letters from memory, you might recall fragments of the sequence such as "DF" or "JK". By parsing the sequence of letters only once, you have already detected frequently occurring patterns and memorized them together as units, i.e. *chunks*. Chunking has been observed in a range of sensory and behavioral modalities including language learning [1, 2], action organization [3, 4] and visual perception of structures [5–7]. Chunking as a mechanism is a basis for humans to identify patterns as objects, assigning labels to them to facilitate memory compression [8, 9], sequence prediction [10, 11], communication [12, 13], and generalization[14]. Learning hierarchical representations of the world is a feature central to human intelligence.

Despite recent success, deep learning models, on the other hand, do not represent explicit hierarchies. Neural networks contain sub-symbolic, nested, non-linear structures whose prediction processes are hard to comprehend. This lack of interpretability raises concerns over their fairness, privacy, robustness and trust-worthiness [15–18] and manifests itself as a key shortcoming of these models [19, 20]. To address these shortcomings, researchers have urged to seek inspiration from cognitive science to construct models that resemble the hierarchical and interpretable representations as observed in human learners [21, 22]. We take a two-fold approach to this problem. First, instead of learning from iid data, we ask: what if the time series data that comprises streams of perception comes from a hierarchical structure? Under this assumption, what could be an algorithm that learns the embedded hierarchical structure? We take inspiration from models in cognitive science showing that people perceive structures based on the Gestalt principle of *grouping by proximity*, and formulate a generic hierarchical pattern discovery algorithm that enables the rational discovery of structures with embedded hierarchies. We refer to this model as the hierarchical chunking model (HCM).

HCM starts out learning a minimal set of units sufficient to explain the sequence and gradually combines these units into increasingly larger and more complex chunks, constructing interpretable hierarchical structures. We derive learning guarantees on an idealized generative model and demonstrate convergence on sequential data coming from this generative model. Thereby, Gestalt principles of grouping can be understood as a rational way of learning representations from sequences with an inherent hierarchical structure. We then show that HCM resembles more to human chunking in qualitative ways compared to a recurrent neural network and flexibly transfers components learned from one task to another. We extend HCM to the visual-temporal domain capable of learning visual-temporal parts and wholes from higher dimensional sequential data. Taking it one step further, we deploy HCM to learn from high-dimensional fMRI data, which exerts a hierarchical structure. We demonstrate HCM's interpretable feature extraction ability to discover submodules of brain activations directly linkable to behavior supported by the literature.

## 2 Hierarchical Chunking Model

We define a chunk as a unit created by concatenating several atomic sequential units together. Taking the training sequence shown in Figure 1a as an example, the sequence is made up of discrete atomic units from an atomic alphabet set $\mathbb{A}_0$: in this case $\mathbb{A}_0 = \{0, 1, 2\}$. A chunk $c$ is made up of a combination of one or more atomic units in $\mathbb{A}_0 \backslash \{0\}$. 0 denotes an empty observation in the sequence.

Intuitively, if a sequence contains inherent hierarchical structure, then there are patterns which span several sequential units sharing these internal structures, examples of such sequences are repeated melodies and sub-melodies in music. If the pattern occurs in the sequence, observations between sequential units within the pattern will be correlated. In this case, chunking patterns within a sequence as units simplifies perceptual processing in the sense that the sequence can be perceived one chunk after another, instead of one sequential unit at a time. Furthermore, the acquired "primary" chunks serve as building blocks to discover larger chunks that are embedded within the hierarchy of the sequential structure.

Formally, HCM acquires a belief set $\mathbb{B}$ of chunks, and uses chunks from the belief set to parse the sequence. HCM assumes that a sequence is generated from samples of independently occurring chunks with probability of $P_{\mathbb{B}}(c)$ evaluated on the belief set $\mathbb{B}$. The probability of observing a sequence of parsed chunks $c_1, c_2, ..., c_N$ can be denoted as $P(c_1, c_2, ..., c_N) = \prod_{c_i \in \mathbb{B}} P_{\mathbb{B}}(c_i)$. Chunks as perceiving units serve as independent factors that disentangle observations in the sequence.

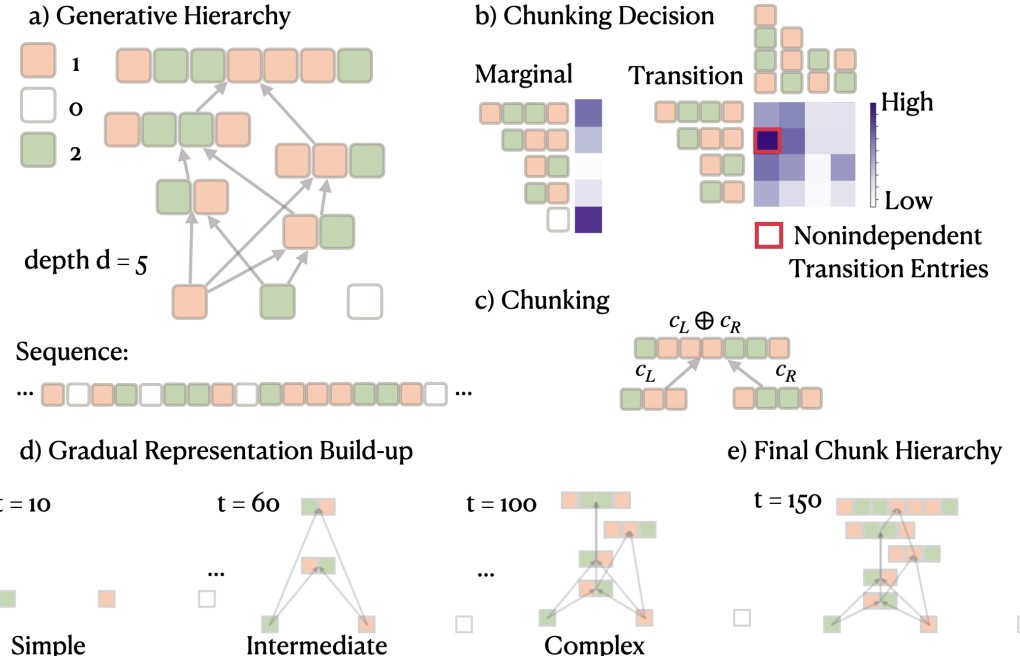

Figure 1: The Hierarchical Chunking Model. **a)** Example of a hierarchical model generating training sequences. **b)** Intermediate representation of learned marginal and transition matrices. The most frequent transition that violates the testing criterion is marked in red and can be turned into a new chunk. **c)** HCM combines the two chunks $c_L$ and $c_R$ to form a new chunk. **d)** As HCM observes longer sequences, it gradually learns a hierarchical representation of chunks. **e)** HCM arrives at the finally chunk hierarchy isomorphic to the generative hierarchy.

The training sequence is parsed by HCM in chunks. At every parsing step, the longest chunk in the belief set consistent with the upcoming sequence is chosen to explain the up-coming sequential observations. The end of the previous parse initiates the next parse.

Observing a hierarchically structured sequence as illustrated in Figure 1a, HCM gradually builds up a hierarchy of chunks starting from an empty belief set. It first identifies a set of atomic chunks to construct its initial belief set $\mathbb{B}$. Initially, these will be chunks of length one, yielding one-by-one processing of the primitive elements.

For one belief set $\mathbb{B}$, HCM keeps track of the marginal parsing frequency $\boldsymbol{M}(c_i)$ for each chunk $c_i$ in $\mathbb{B}$, a vector with size $|\mathbb{B}|$ and the transition frequency $\boldsymbol{T}$ between chunk $c_i$ followed by chunk $c_j$, as illustrated in Figure 1b. Entries in $\boldsymbol{M}$ and $\boldsymbol{T}$ are used to test the hypothesis that consecutive chunk parses have a correlated consecutive occurrence within the sequence via a $\chi^2$-independence test. If two chunks $c_L$ and $c_R$ have a significant adjacency dependence based on their entries in $\boldsymbol{M}$ and $\boldsymbol{T}$, they are chunked together to become $c_L \oplus c_R$, which augments the belief set $\mathbb{B}$ by one. One example of chunk merging is shown in Figure 1c.

**Independence Test** We use a $\chi^2$-test of independence to assess the correlation of consecutive occurrences of $c_L$ followed by $c_R$. Let $c_L$ be an indicator variable that is 1 when chunk $c_l$ is parsed and 0 otherwise, similarly we formulate $c_R$ as another indicator variable of parsing the chunk $c_r$. We evaluate the $\chi^2$-value as a criterion to reject the null hypothesis that the consecutive observation of $c_l$ followed by $c_r$ is statistically independent:

$$\chi^2 = \sum_{c_L=\{0,1\}} \sum_{c_R=\{0,1\}} \frac{N(p(c_L, c_R) - p(c_L)p(c_R))^2}{p(c_L)p(c_R)}$$

$p(c_L, c_R)$ and $p(c_L)p(c_R)$ are evaluated from $\boldsymbol{M}$ an $\boldsymbol{T}$. The degree of freedom is 1. A $\chi^2$-probability of less than 0.05 is the criterion to reject the null hypothesis (i.e. that $c_l$ and $c_r$ occur independently).

There are two versions of HCM. The Rational Chunk Learning HCM learns chunks in an idealized way which we use to study learning guarantees. The online version of HCM is an approximation

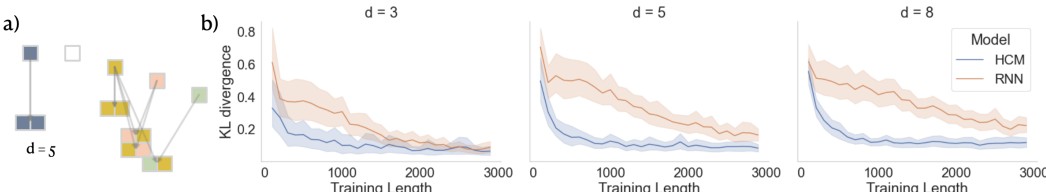

Figure 2: **a)** Example graph generated from the hierarchical generative model with a depth of $d = 5$. **b)** Learning performance of HCM and RNN with increasing training length and for increasing depths. Performance was averaged over 30 randomly-generated graphs.

to the rational HCM that can be adapted to different environments and processes sequences online. Pseudo-code for both algorithms can be found in the SI.

**Rational Chunk Learning: HCM as an Ideal Observer** HCM is initiated with an empty belief set and it first finds a minimally complete belief set after the first sequence parse. In each iteration, the entire sequence is parsed to evaluate $M$ and $T$, which are used to find consecutive chunk parses in the existing belief set that violate the independence testing criterion. From these dependent chunk pairs, the pair with the largest estimated joint probability is combined into a new chunk. The new chunk enlarges the belief by one. The chunks in the new belief set are used to parse the sequence in the next iteration. This process repeats until all of the chunks in the belief set pass an independence halting criterion, which measures if all of the chunks in the belief set are currently independent, again assessed via a $\chi^2$-test (see SI).

**Online Chunk Learning** The online chunk learning HCM approximates the ideal observer HCM by learning new chunks when the training sequence is processed on the go. To have a feature that encourages adaptation to new environmental statistics, entries in $M$ and $T$ can be subject to memory decay. We use the ideal observer model to demonstrate learning guarantees, but use the online model to learn representations in realistic and more complex set-ups.

## 2.1 HCM Learns Representations from the Ground Up

As HCM learns from a sequence, it starts with no representation and gradually builds up interpretable representations described by a chunk hierarchy graph $\hat{\mathcal{G}}$ with the vertex set being the chunks and edges pointing from constituents to composites. Shown in Figure 1d is the gradual build-up of one such graph as the model learns from a training sequence coming from the generative hierarchy in Figure 1a. At $t = 10$, HCM learns only the atomic chunks, at $t = 60$, HCM has already constructed two additional chunks; when $t = 100$, two more additional chunks are constructed. HCM arrives at the final chunk hierarchy at $t = 150$.

## 3 Generating Sequences with a Hierarchical Structure

We construct a generative model to study HCM's behavior formally and empirically. The generative model constructs random chunk hierarchies from which non-iid sequences are sampled. Such graph $\mathcal{G}_d$ contains vertex set $V_{\mathbb{A}_d}$ and edge set $E_{\mathbb{A}_d}$ to describe the relation between chunks and their constituents. One example is illustrated in Figure 1a. $\mathbb{A}_d$ is the set of chunks used to construct the sequence. The depth $d$ specifies the number of chunks created in the generative process.

Starting with an initial set of atomic chunks $\mathbb{A}_0$, at the i-th iteration, two chunks $c_L$, $c_R$ are randomly chosen from the current set of chunks $\mathbb{A}_i$ and are concatenated into a new chunk $c_L \oplus c_R$, augmenting $\mathbb{A}_i$ by one to $\mathbb{A}_{i+1}$. Meanwhile, an independent occurrence probability is assigned to each chunk under the constraint that the probability of occurrence for every new chunk $c_i$ in the construction process evaluated on the support set $\mathbb{A}_i$ carries the largest probability mass.

Once a graph hierarchy is constructed, we construct non-iid observational sequences by consecutively sampling chunks from the hierarchy with their corresponding probability, under the constraint that no two chunks with a child chunk are sampled consecutively.

### 3.1 Learning Guarantee

**Theorem**: As the length of the sequence approaches infinity, HCM learns a hierarchical chunking graph $\hat{\mathcal{G}}$ isomorphic to the generative hierarchical graph $\mathcal{G}$.

*Proof Sketch*: We approach this proof by induction. Further details can be found in the SI. Base step: The first step of the rational chunking algorithm is to find the minimally complete atomic set of chunks to form its initial belief set. This procedure guarantees that $\hat{\mathcal{G}}_0 = \mathcal{G}_0$. Additionally, the probability mass of the learning model at step 0 and the generative model at step 0 is asymptotically the same as the sequence length approaches infinity. Induction hypothesis: Assume that the learned belief set $\mathbb{B}_i$ at step $i$ contains the same chunks as the alphabet set $\mathbb{A}_i$ in the generative model, the chunk combination pair with the biggest evaluated joint occurrence probability violating the independence test is picked to be concatenated into a chunk to extend the belief set: this chunk is the same chunk node created by the hierarchical generative model. End step: The chunk learning process stops once the independence criterion is no longer violated. This is the case once the chunk learning algorithm has learned a belief set $\mathbb{B}_d = \mathbb{A}_d$.

### 3.2 Learning Convergence and Comparative Data-efficiency

To evaluate and show HCM's learning performance, we trained HCM to learn hierarchies of chunks from sequences generated by the hierarchical generative model. Shown in Figure 2 is HCM's learning performance as sequence length increases, averaged over 30 independently generated random graphs with the same depth $d$. One example of such graphs is shown in Figure 2a. Kullback-Leibler divergence was used to evaluate learning performance. To this end, learned hierarchies by HCM were used to generate sequences, which were then evaluated on the support set of the alphabet set in the generative model.

Figure 2b shows the KL-divergence between the learned and ground-truth distribution for increasing depths $d$ of the generative graphs. For each depth, the KL-divergence was evaluated on 30 random generative models with sequence length increasing from 50 to 3000. Overall, the KL-divergence decreased as the training sequence length increased and converged with longer training sequences, showing a closer representation resemblance to the generative model.

A similar training and learning evaluation was conducted on a 3-layer Recurrent Neural Network (RNN) with 40 hidden units for comparison. As the length of the training sequence increased, the KL-divergence of RNN converged at a slower rate than HCM. This competitive advantage in data efficiency became more pronounced with increasing depth of the generative hierarchy.

## 4 HCM Resembles Human Chunk Learning

Here we compare the chunk learning behavior of HCM to the learning characteristics of humans. To that end, we used data collected from a sequence learning study by [23] with 47 participants under the license CC-BY 4.0. As shown in Figure 3a, the training sequence comprised chunks ABC and D, independently occurring with equal probability. The study assessed how participants built up chunk knowledge gradually. Participants' reaction times reflected that, after enough training, they were anticipating several upcoming sequence elements, suggesting that they have acquired longer chunks (Figure 3b, left) [23].

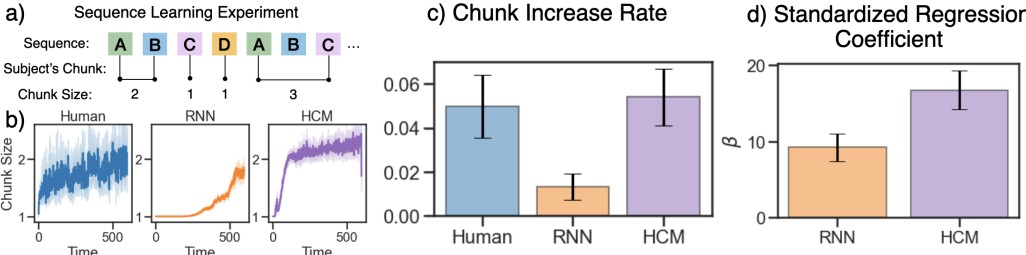

Figure 3: **a)** A sequence learning experiment with chunks ABC and D. **b)** Chunk size increase of human participants, RNN, and HCM during training. **c)** Average chunk increase rate during training. **d)** Regression coefficient of RNN and HCM's confidence estimates on human reaction time data.

In a similar vein, we measured online chunk size increase of HCM using the same method as in [23] and, for comparison, RNNs (see SI for further comparisons to other algorithms). HCM, similarly to humans, started learning longer chunks early in the sequence. By contrast, RNN did not start to chunk until after step 300, and when it started to learn chunks, the increase rate of the predictive horizon was not as steep as HCM's. Evaluating the average rate of chunk growth also showed that HCM builds up chunks as learning progresses was more similar to participants' than the RNN's (Figure 3c). The negative log-probabilities of sequence elements generated by the HCM and RNN were both significantly related to human reaction times (that reflect the certainty of their internal predictions [24]). Yet, the relationship was substantially stronger (Figure 3d) between HCM ($\beta = 16.74$, $p \leq 0.001$, $\tau = 0.165$, $BIC = 313586.5$) and human participants compared to that of the RNN ($\beta = 9.24$, $p \leq 0.001$, $\tau = 0.085$, $BIC = 314236.4$). These results suggest that HCM resembles human chunk learning more strongly than RNNs and can therefore be seen as the cognitively more plausible approach to hierarchical representation learning.

## 5 HCM Permits Transfer Between Environments

One characteristic of human learning is that previous learning experience facilitates and sometimes interferes with acquiring a new skill [25]. Having an interpretable representation can inform us about positive or negative transfer a priori.

An HCM has learned a chunking graph in Figure 4a from an environment. When it switches to another environment with a generative model overlapping with its previously acquired representation, it can reuse the learned subgraphs of chunks marked in gray as in Figure 4b and learns faster than a naive HCM in Figure 4c. Vice versa, transfer can be detrimental when there is no or little overlap between the learned chunking graph and the generative model of the new environment. For the same HCM as in Figure 4a, transferring to an environment with a chunking graph in Figure 4d implies learning the shaded chunks anew, in addition to running the risk of being misled by the previous representations. As a result, the early performance of the pre-trained HCM suffers more from an interfering environment than a naive model in Figure 4e. Interpretability of HCM's representations enables the assessment of facilitation or interference when the environment changes.

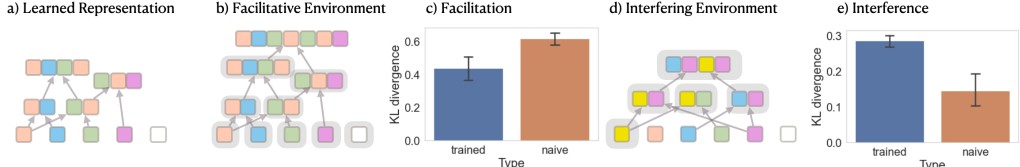

Figure 4: **a)** Example of a representation learned by an HCM. **b)** A facilitative environment with a generative model overlapping in its structure with the test environment. Gray shadows mark the chunks that can be directly transferred. **c)** Average performance over the first 500 trials after the environment switches. **d)** Interfering environment. Gray shadows marks chunks that need to be acquired anew. **e)** Average performance over the first 500 trials after the environment switches.

## 6 Generalizing to Visual Temporal Chunks via the Principle of Proximal Grouping

Humans excel at finding structures in hierarchical visual objects and grouped movements. The Gestalt principle of *grouping by proximity* suggests that we tend to group objects that are close to one another into a cohesive unit [26, 27]. This principle has been suggested to play a key role in human perceptual grouping [28], benefits working memory [29] and reduces visual complexity [30]. Indeed, in humans and other animals, learning of adjacent relationships prevails over non-adjacent ones [31]. Therefore, the adjacent dependency structure can be expanded to chunking in visual temporal domains [32]. To emulate this ability of chunking via proximal grouping, we extend HCM to learn visual temporal chunks.

Visual temporal chunks subsume temporal length and varying visual slices in each temporal slice (Figure 5a). One can imagine a visual temporal chunk as having a 3D shape — the first two

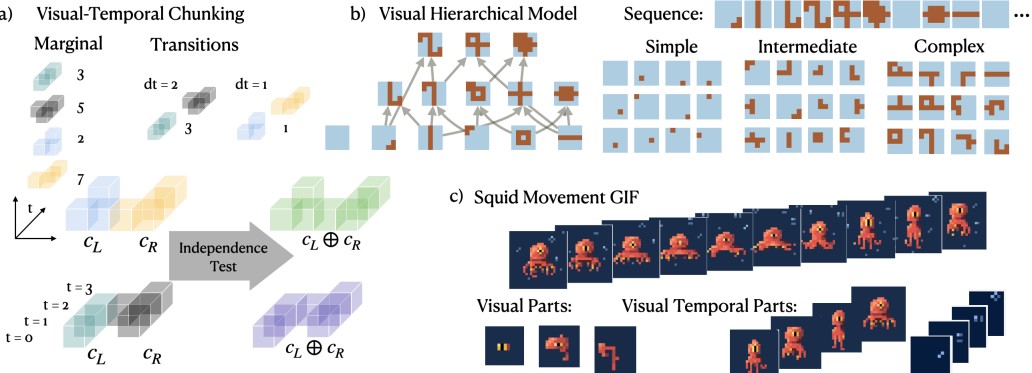

Figure 5: **a)** HCM learns visual-temporal chunks by extending the transition matrix to take account of time differences. **b)** Left: A visual hierarchical model where complex images are composed of simpler images. Right: Initial, intermediate, and complex chunks learned by HCM trained on sequences of images sampled from the visual hierarchical model.**c)** Top: A GIF of a moving visual used as a sequence to train HCM. Bottom: Examples of temporal and visual chunks learned by HCM.

dimensions are the visual part of the chunk, and the object's length is the temporal part, made of stacked visual-temporal pixels. Within each temporal slice are the visual features identified by the chunk. As the model iterates through data across its temporal slice, the chunk that attains the biggest visual temporal volume explains part of the observational sequence. Multiple visual temporal chunks can occur simultaneously. Starting at the visual temporal time point marked by the previous chunk, chunks are identified and stored in $M$. The transition matrix $T$ is modified to account for the temporal lag difference between adjacent chunk pairs within a proximity parameter and records the frequency that one chunk transitions into another for each time lag. Whenever a pair of adjacent chunks are identified, an independence hypothesis test evaluates whether the adjacent observation are correlated. Chunks that violate the hypothesis test are combined to parse future sequences.

**Learning Part-Whole Relationship Between Visual Components**    We show HCM learned chunks in the visual domain from a sequence of independently sampled images. Figure 5b left shows a hierarchical generative model in the pixel-wise image domain. A set of elementary visual units in the lowest hierarchy level combines into intermediate and more complex visual units higher up in the hierarchy. All of the constructed elements in the hierarchy occurred independently according to a probability drawn from a Dirichlet flat distribution. Images in the hierarchy were independently sampled from the generative distribution to become the training sequence. In Figure 5b, right we show the chunk representations learned by HCM at different stages. Initially, HCM acquires the individual pixels as chunks to explain the observations. As HCM proceeds with learning, it discovers visual correlations among the pixels and constructs increasingly complex visual parts.

**Learning Visual-Temporal Movement Hierarchies**    Instead of seeing one image after another sampled from an independent, identically distributed distribution, real-world experiences contain correlations in both the visual and temporal dimensions. From observing object movements across space and time, the visual system learns structures from correlated visual and temporal observations, decomposes motion structure, and groups moving objects together as a whole [33]. To emulate this type of environment, an animated GIF of a squid swimming in the sea (Figure 5c) was used as a visual-temporal sequence to train HCM. As learning advances, HCM learns chunks spanning both the visual and temporal domains. There are visual-temporal chunks that mark the movements of a tentacle and the rising-up motion of a bubble. Additionally, visual chunks resemble a part of the visual's eye and face. The meaningful chunks in the visual-temporal domain suggest the grouping principle enables the plausible learning of movement sequences and aids the perception of objects as wholes and their corresponding parts.

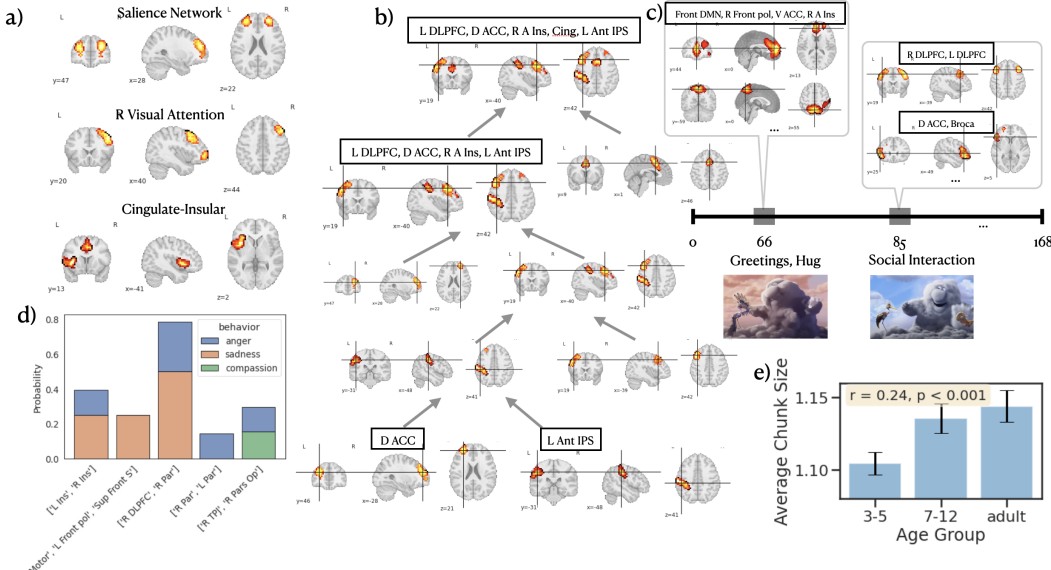

Figure 6: Application avenues of HCM on fMRI data. **a)** Example chunks of brain functional activation regions. **b)** HCM learns hierarchical functional network with bigger chunks emerging from its constituents. **c)** Chunk activation patterns responding to scene content **d)** Distinct response of retrieved chunks to tagged scenes. **e)** Average chunk size across age groups.

# 7   Learning Hierarchies of Brain Activation from Resting-state fMRI data

HCM learns hierarchies from structured sequential data. As brain activation has been suggested to be hierarchically structured [34], we demonstrate HCM's usefulness to learn structures in biological neural networks activating in response to complex stimuli by running HCM on a resting-state fMRI data set.

We used a developmental data set provided by the `nilearn` package with BSD License [35] and originally collected by [36] with its corresponding IRB approval. This data set contains the resting-state BOLD activity of 155 participants ranging from age 3 to 40, while watching the silent movie "Partly Cloudy". BOLD signal was extracted from functional brain regions defined by the MSDL Atlas [37], with confounds excluded and transformed into a rounded, normalized time series.

**HCM's Chunks Reflect Structural, Functional and Anatomical Connectivity**    Figure 6 shows three typical examples of learned chunks for a randomly-chosen participant. The labels of functional regions come from the MSDL atlas [37]. The first example is the co-activation of D ACC and R A Ins. These two regions have been observed to co-activate in the presence of emotions, pain, and humor. They have been suggested to be a key hub of the salience network [38–41]. The second example chunk contains the activation of R DLPFC and R Front Pole. These regions belong to the visual attention network and are known to be anatomically connected [42–44]. A final example is the chunk of L Ins and Cing, which are also known to be anatomically and functionally connected [45]. Thus, the chunks discovered by HCM correspond to empirically-verified patterns of functional activity.

**HCM's Chunks Recover Hierarchical Activation Patterns**    In fMRI data, hierarchies of chunk activation constructed by HCM reflect networks of functional regions. On the top of the hierarchy, the largest chunk contained L DLPFC, D ACC, R A Ins, Cing, and L Ant IPS (Figure 6b). Those regions are known to co-activate during cognitive tasks that demand attention, working memory, and control [41]. Chunks in the intermediate levels of the hierarchy reflect sub-networks of functional connectivity. Sub-chunks such as D ACC, R A Ins, L Ant IPS, and L DLPFC have been suggested to conjointly activate in cognitive effort-related activities [41]. Atomic chunks in the hierarchy such as D ACC and L Ant IPS activate individually sometimes without their parent chunks. Indeed, they have distinct functional signatures for affect processing [46, 47], and visual attention control [48]. Upon exposure to a time series in fMRI data, HCM constructs chunks from their constituents and arrives at a hierarchy of chunk relations, indicating nested network structures in the human brain.

**HCM's Chunks can be Matched with Stimulus Onsets**    The retrieved chunks by HCM can be tagged with critical stimulus onsets. We tagged 19 critical moments involving social and emotional content in the movie. Figure 6c shows one example chunk activation upon stimulus onsets. Frontal DMN, right frontal pole, ventral anterior cingulate cortex, and right anterior insula activate together as a recurring unit after participants witness a scene with characters greeting and hugging each other. These regions have been suggested to be involved in social and cognitive processing [49]. Another example is the activation of areas known to be involved in emotion and language processing: D ACC and Broca [50], during a scene containing social interactions. In the meantime, the left and right prefrontal cortex, involved in theory-of-mind [36], also lights up.

We categorized the tagged moments into 3 groups of different emotional load: sadness, anger, and compassion. We then looked at the activation probability of retrieved chunks within the 6 seconds after watching those tagged scenes. Figure 6d shows a list of such chunks from one participant with their activation probability for each emotional category. For example, the left and right insula, known to be involved in affective processing [51], have a 0.4 activation probability after witnessing scenes of sadness or anger, but no activation after witnessing scenes of compassion. The same holds for R DLPFC and R Par that have been documented to activate in response to emotional conflict [52]. On the other hand, regions such as R TPJ and R pars opercularis that are involved in emotional reactions and theory of mind processes [53], activate in response to a scene of compassion or anger, but not to a scene of sadness. Thus, chunks of active brain regions can be related to complex stimuli, and regions activate selectively in response to one or more categories of emotional stimuli, but not others.

**HCM's Average Chunk size Correlates with Participants' Age**    HCM can also be used to perform meaningful analyses at the population level. Specifically, we find that HCM's returned average chunk size per participant correlates significantly with age (Figure 6e). The older the participants are, the longer are the chunks found in their data ($r = 0.23$, $p \leq 0.001$). This discovery is in line with findings in the original study, which showed an increase in modularization of ToM and pain circuits across development [36].

To summarize, we applied HCM to learn chunks from a developmental fMRI data set. HCM enabled the discovery of spatially and temporally correlated activation chunks that are theoretically and empirically meaningful. The resulting chunks can be linked to complex stimuli and offer directly interpretable insights into the structure and function of brain activity.

## 8    Related Work

HCM extends upon decades of previous cognitive science and psychology research on chunking. In cognitive science, process models such as PARSER and competitive chunking were demonstrated to generate qualitatively similar chunks as in human sequence learning [54, 55]. HCM is a rational algorithm that learns the underlying chunks when the sequence is generated from a hierarchical chunking graph. Therefore, the chunking criterion is no longer a heuristic but a rational learning strategy that enables hierarchical structural discovery. On top of inheriting the merits of its predecessors, HCM generalizes the chunk learning principle to higher dimensional sequential domains such as visual-temporal sequential data.

HCM relates to several other lines of research. One is program induction. In program induction, explicit representations are acquired by searching for programmatic structures that best explain observational samples [22], and consolidating these offline with library learning [56]. However, domain expert knowledge is needed to specify the primitive programs; the relations and composition rules must adapt to the task settings and sensitively influences the quality of retrieved representations. Other approaches to structure learning include unsupervised parsing [57], which learns a stochastic and-or graph from sequential data. HCM is distinct in adapting its representation granularity to discover bigger chunks from data without pre-specifying the structure.

Another category of models to learn from sequential data are traditional sequence learning models including Hidden Markov Models (HMM), n-gram models and their variants to capture multi-scale sequential structure such as Hidden semi-Markov model [58] and hierarchical HMM [59]. The parameters of these models proliferate exponentially as a function of chunk length, implying memory inefficiency. Additionally, these models demand a structure specification before fitting parameters to the data. They also lack the adaptive recombination and reuse of pre-existing components. The

same issue is with neural network approaches to extract chunks from sequences [60–62]. Apart from lacking in interpretability, these models do not leverage the concatenation process observed in humans or reusing previously learned representations to construct new representations.

The principle of iterative merging of chunks has been used in compression algorithms, such as tokenizing methods in NLP, which were developed to optimize sequential data compression. Tokenizing methods such as Byte-Pair Encoding [63] iteratively merges the most frequent pairs of chunks to build a vocabulary of a text corpus. This objective is easy to compute but gives rise to ambiguous parses of the text (e.g. [AB, C] / [A, BC]). To minimize parse ambiguity, WordPiece [64] merges chunks that increase the likelihood of the corpus the most. However, the objective of WordPiece is expensive to compute. HCM circumvents the problem of computing the global sequence likelihood by instead maximizing the local chunk continuation likelihood. The computational efficiency of HCM makes it a plausible cognitive model of chunking as well as a promising method for NLP.

Probabilistic context-free grammars (PCFGs) are related to HCM in that they use trees as a representational form of sequences. The parse trees of PCFGs denote production rules, such as S → NP + VP. These production rules define how abstract syntactic units (non-terminals), such as a noun phrase and a verb phrase, are instantiated into a concrete string of words (terminals) to compose a sentence, such as 'we wrote the paper'. In comparison, the generative tree of HCM denotes statistical relationships among concrete chunks, such as 'we'-'wrote'-'the paper'. Extending HCM to represent abstraction is an exciting future direction, on which avenue the comparison to PCFGs will be instructive.

## 9  Discussion

Our work has its limitations. Currently, we fix the memory decay and the deletion threshold parameters to a priori plausible values. In future work, these parameters could be adapted online based on environment volatility. Another limitation is its scalability: at the moment, HCM learns representations from semi-high dimensional sequential data (i.e., currently between 1 to 625 dimensions). We are actively looking into generalizing this algorithm to higher dimensional data domains by combining it with existing neural network approaches or computer vision algorithms such as coherent point drift [65] or normalized cuts [66] to allow for the learning of ambiguous and high dimensional chunk exemplars. It is also possible to combine HCM with the compressed representation, such as the hidden activity of an auto-encoder to process and learn the structure from downstream representation. In this work, HCM learns one type of hierarchy of compound representations. However, we can show that HCM can be generalized to not only learn simple chunks but also chunks in projected spaces and thereby generalize between two chunks that contain the same motif (for example, "12221212" and "34443434"; see SI for detailed results). In the future, it might be worthwhile to further combine our approach with others amongst a taxonomy of representational hierarchies.

HCM also opens up other application directions. One direction is integrating HCM with deep neural network approaches as an interface between human understanding and distributed computation. Learning hierarchies of coherent activations from intermediate hidden units has the potential to reveal neural networks' underlying computation structure. Furthermore, it is also possible to equip HCM with additional top-down encoded representations, for example, by pre-training on other sequences or by adjusting the chunks by hand before the training starts. Another direction in neuroscience or behavioral research is to learn chunks of tagged animal movements that enable insights into the emergence of behavioral structure [67]. Finally, finding patterns that form as a cognitive unit is a vital task for infants to learn about the structures of the world and resembles the process of formulating a scientific theory from observation [68]. HCM can function as one means to come up with world models by observation, ready for experimental interventions or active learning to delineate the causal structure within [69].

## 10  Conclusion

We have proposed a hierarchical chunking model (HCM) that learns chunks from non-iid sequential data with a hierarchical structure. HCM starts out learning an atomic set of chunks to explain the sequence and gradually combines them into increasingly larger and more complex chunks. The output of the model is a dynamical graph that is a trace of the evolving representation. The resulting representations are easy to interpret, and flexibly reusable.

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
