# Supplementary Information: Learning Structure from the Ground up—Hierarchical Representation Learning by Chunking

**Shuchen Wu**[*]
Computational Principles of Intelligence Lab
Max Planck Institute for Biological Cybernetics
Tübingen, Germany
shuchen.wu@tuebingen.mpg.de

**Noémi Éltető**
Department of Computational Neuroscience
Max Planck Institute for Biological Cybernetics
Tübingen, Germany
noemi.elteto@tuebingen.mpg.de

**Ishita Dasgupta**
Computational Cognitive Science Lab
Department of Psychology
Princeton University
dasgupta.ishita@gmail.com

**Eric Schulz**
Computational Principles of Intelligence Lab
Max Planck Institute for Biological Cybernetics
Tübingen, Germany
eric.schulz@tuebingen.mpg.de

## A   Definitions

An observational sequence is made up of discrete, integer valued, size-one elementary observational unit coming from an atomic alphabet set $\mathbb{A}_0$, where 0 represents the empty observational unit.

One example of such an observational sequence $S$ is:

$$010021002112000...$$

The atomic alphabet set is $\mathbb{A}_0 = \{0, 1, 2\}$. The elementary observation units are '0', '1', and '2'.

**Definition 1 (*Chunk*)**

*A chunk is made from any combinations of non-empty observational units $\mathbb{A}_0 \setminus \{0\}$.*

Examples of chunks from the observational sequence can be '1', '21', '211', '12', '2112', ... etc. 0 represents an empty observation in the sequence.

**Definition 2 (*Belief Set*)**

*A belief set is the set of chunks that HCM uses to parse sequences, denoted as $\mathbb{B}$.*

An example belief set that HCM has learned to parse sequence $S$ can be: $\mathbb{B} = \{0, 1, 21, 211, 12, 2112\}$.

---

[*]

**Definition 3 (*Parsing*)**

*Chunks are being parsed from the beginning of the sequence. At each parsing step, the biggest chunk in the belief set that matches the upcoming sequence is chosen to explain the observation. The end of the previous parse initiates the next parse.*

Using the belief set $\{0, 1, 21, 211, 12, 2112\}$ to parse the sequence $S$ results in the following partition. $\underline{0}\ \underline{1}\ \underline{0}\ \underline{0}\ \underline{21}\ \underline{0}\ \underline{0}\ \underline{2112}\ \underline{0}\ \underline{0}\ \underline{0}$.

**Definition 4 (*Completeness*)**

*We say that a belief set is* complete *if at any point when the model parses the sequence, the upcoming observations can be explained by at least one chunk in the belief set.*

In this work, we only refer to complete belief sets.

**Definition 5 (*Parsing Length $N_{\mathbb{B}}$*)**

*A parsing length $N_{\mathbb{B}}$ of a sequence is the length of resulting sequence after being parsed by chunks in $\mathbb{B}$.*

**Definition 6 ($N_{\mathbb{B}}(c)$)**

*$N_{\mathbb{B}}(c)$ denotes number of times chunk $c$ in the belief set $\mathbb{B}$ appears in the parsed sequence.*

$N_{\mathbb{B}}(c)$ for all of the chunks $c$ on a belief set $\mathbb{B}$ sums to the parsing length: $N_{\mathbb{B}} = \sum_{c \in \mathbb{B}} N_{\mathbb{B}}(c)$

**Definition 7 ($N_{\mathbb{B}}(x \to y)$)**

*The number of times chunk $x$ is being parsed following chunk $y$. $x$ and $y$ are both chunks in the belief set $\mathbb{B}$.*

For any chunk $x$ within any belief set $\mathbb{B}$, $N_{\mathbb{B}}(x)$ has the following relation with $N_{\mathbb{B}}(x \to y)$:

$$N_{\mathbb{B}}(x) = \sum_{y \in \mathbb{B}} N_{\mathbb{B}}(x \to y) \tag{1}$$

When the length of the sequence becomes infinite, it is easier to work with probabilities instead of counting the number of chunk occurrences.

**Definition 8 (*Probability space of a belief set*)**

*With a belief set $\mathbb{B}$, one can define a associated probability space $(\mathcal{S}_{\mathbb{B}}, \mathcal{F}_{\mathbb{B}}, \mathcal{P}_{\mathbb{B}})$. $\mathcal{S}_{\mathbb{B}}$ is the sample space representing all of the possible outcomes of a chunk parse. An event space $\mathcal{F}$ is the space for all possible sets of events. $\mathcal{F}$ contains all the subsets of $\mathcal{S}_{\mathbb{B}}$. Additionally, the probability function $P_{A_{\mathbb{B}}} : \mathcal{F}_{\mathbb{B}} \to \mathbb{R}$ is defined on the event space $\mathcal{S}_{\mathbb{B}}$. The probability function $P_{A_{\mathbb{B}}}$ satisfies the basic axioms of probability:*

- *$P_{A_{\mathbb{B}}}(E) \geq 0\ \forall E \in \mathcal{F}$. For any subset in the event space, the probability of an observation being in the subset is positive.*
- *$M, N \in \mathcal{F}$, and $M \cap N = \mathbb{E}$, then $P(M \cup N) = P(M) + P(N)$. For two non-intersecting subsets in the event space, the probability of observing any element that falls within the union of the two subsets is the sum of the probability of observing any event within one subset and the probability of observing any event from the other subset.*
- *$P(S) = 1$. The probability of observing any event that belongs to the sample space is one.*

In the limiting case when the sequence becomes infinitely long, we formulate the probability of parsing chunk $c \in \mathbb{B}$.

$$P_{\mathbb{B}}(c) = \lim_{N_{\mathbb{B}} \to \infty} \frac{N_{\mathbb{B}}(c)}{N_{\mathbb{B}}} \tag{2}$$

A learning model keeps track of the occurrence probability associated with each chunk in the belief set. For a current belief set, the model assumes that the chunks within the belief set occurs independently.

The probability of observing a sequence of chunks $c_1, c_2, ....c_N$ can be denoted as $P(c_1, c_2, ....c_N)$. The joint probability of observing any chunk in the generative process is:

$$P(c_1, c_2, ....c_N) = \prod_{c_i \in \mathbb{B}_d} P_{\mathbb{B}_d}(c_i) \tag{3}$$

Chunks as observation units serve as independent factors that disentangle observations in the sequence.

**Definition 9 (*Marginal Parsing Frequency $M_d$*)**

*A vector that stores the number of parses for each chunk $c$ in the belief set $\mathbb{B}_d$.*

$M_d$ is a vector with size $|\mathbb{B}_d|$.

**Definition 10 (*Transition Frequency $T_d$*)**

*The set of transition frequency from any chunk $c_i \in \mathbb{B}_d$ to $c_j \in \mathbb{B}_d$*

**Definition 11 (*Chunk Hierarchy Graph $\mathcal{G}_d$*)**

*The relation between chunks and their constructive components in the generative model is described by a chunk hierarchy graph $\mathcal{G}_d$ with vertex set $V_{\mathbb{A}_d}$ and edge set $E_{\mathbb{A}_d}$. In this hierarchical generative model, $d$ is the depth of the graph and $\mathbb{A}_d$ is the set of chunks used as atomic units to construct the sequence. Each vertex in $V_{\mathbb{A}_d}$ is a chunk, and edges connect the parent chunk vertices to their child chunk vertices.*

# B   Independence Test as a Chunking Criterion

Combining any two chunks $c_L$ and $c_R$ in the current belief set by ranking their joint occurrence probability may result in combining independently occurring chunks together. To distinguish this scenario of taking precedence over correlated and yet lower probability chunk pairs, we use Pearson's chi-square statistic for evaluating statistical independence to assess if the consecutive parses of $c_l$ and $c_r$ observed in $T$ are independent. We use a $\chi^2$-test of independence to assess the correlation of consecutive occurrences of $c_L$ followed by $c_R$ in $\mathbb{B}$. Let $c_L$ be an indicator variable that is 1 when chunk $c_l$ is parsed and 0 otherwise, similarly we formulate $c_R$ as another indicator variable of parsing the chunk $c_r$. Observations of $c_L$ and $c_R$ in parses are categorical variables and can be represented as rows and columns of a contingency table. The number of observations that $c_L = 1$ or any other observations ($c_L = 0$) consists of the row entries, indicating observations of $c_L$, while the number of observations $c_R = 1$ and $c_R = 0$ make up the column entries. The table, therefore, consists of two rows and two columns.

The null hypothesis is the statistical independence of consecutive observations. Given the independence hypothesis, the expected frequency for observing $c_l$ followed by $c_r$ is $E[c_L, c_R] = Np(c_L)p(c_R)$, with $N$ being the total number of parses.

$$\chi^2 = \sum_{c_L=\{0,1\}} \sum_{c_R=\{0,1\}} \frac{(O(c_L, c_R) - E[c_L, c_R])^2}{E[c_L, c_R]}$$

$$= \sum_{c_L=\{0,1\}} \sum_{c_R=\{0,1\}} \frac{N(p(c_L, c_R) - p(c_L)p(c_R))^2}{p(c_L)p(c_R)} \tag{4}$$

The degree of freedom for this test is 1. A $\chi^2$ value of less than or equal to 0.05 is used as a criterion for rejecting the null hypothesis of independence.

## B.1   Independence Test as a Halting Criterion

In the rational version of the chunking algorithm, the independence test is also employed to evaluate the strength of statistical correlation between chunks in the current belief set as a criterion to continue

or to halt the chunking process. In this case, the contingency table contains rows and columns corresponding to all possible chunks in the current belief set, and the $\chi^2$-statistic is calculated as:

$$\chi^2 = \sum_{c_L \in \mathbb{B}} \sum_{c_R \in \mathbb{B}} \frac{(O(c_L, c_R) - E[c_L, c_R])^2}{E[c_L, c_R]} = N \sum_{c_L \in \mathbb{B}} \sum_{c_R \in \mathbb{B}} \frac{(p(c_L, c_R) - p(c_L)p(c_R))^2}{p(c_L)p(c_R)} \tag{5}$$

The degrees of freedom are $(|\mathbb{B}| - 1)^2$, and a $p$-value of $0.05$ is used as a criterion to reject the null hypothesis and thereby used as an evidence to continue the chunking process.

We chose 0.05 as a decision criterion for rejecting the null hypothesis of two consecutively occurring chunks to be independent to be consistent with standard conventions in statistics. However, in applications, the strictness of parameters can be adapted to task domains. For example, for medical data, you might want to have only a few chunks and should, therefore, set alpha to be conservatively low. However, when using chunks for predictions in downstream tasks, you might want to have more of them, and should, therefore, set alpha to be liberally high.

## C   Rational Chunking Algorithm

---

**Algorithm 1:** Rational Chunking Algorithm

---
**input** : Seq, maxIter
**output** : $\mathbb{B}_d$, $\hat{\mathcal{G}}_d$, $\boldsymbol{T}_d$, $\boldsymbol{M}_d$
$d \leftarrow 0 \ iter \leftarrow 0$;
$\mathbb{B}_d, \boldsymbol{M}_d, \boldsymbol{T}_d = \text{getSingleElementSets(Seq)};$         /* minimally complete atomic set */
**while** *!Test($\boldsymbol{M}_d$, $\boldsymbol{T}_d$) and iter $\leq$ maxiter* **do**
    $\boldsymbol{M}_d, \boldsymbol{T}_d = \text{Parse(Seq}, \mathbb{B}_d)$;
    $c_L, c_R \leftarrow None$;
    $MaxChunk, MaxChunkP \leftarrow None$;
    $PreCk = \{\}$;
    **for** $(c_i, c_j) \in \mathbb{B}_d \backslash \{0\} \times \mathbb{B}_d \backslash \{0\}$ **do**
        $P_d(c_i \oplus c_j) = CalculateJoint(\boldsymbol{M}_d, \boldsymbol{T}_d, c_i, c_j)$;
        $P_{d+1}(c_i \oplus c_j) = \frac{P_d(c_i \oplus c_j)}{1 - P_d(c_i \oplus c_j)}$;
        **if** $P_{d+1}(c_i \oplus c_j) \geq MaxChunkP$ *and* $c_i \oplus c_j \notin PreCk$ *and !Test($c_i$, $c_j$)* **then**
            $c_L \leftarrow c_i, c_R \leftarrow c_j$;
            $MaxChunkP \leftarrow P_{d+1}(c_i \oplus c_j)$;
            $MaxChunk \leftarrow c_i \oplus c_j$
        **end**
    **end**
    $c \leftarrow c_L \oplus c_R$;
    $\mathbb{B}_{d+1} \leftarrow \mathbb{B}_d \cup c$;
    $\hat{\mathcal{G}}_{d+1} \leftarrow \text{AugmentGraph}(\hat{\mathcal{G}}_d, (c_L, c), (c_R, c))$;
    $PreCk.add(c)$;
**end**

---

## D   Online HCM and Generalization to Visual-Temporal Sequences

Online HCM learns a chunk hierarchy graph $\hat{\mathcal{G}}$ from visual-temporal sequences. The chunk hierarchy graph $\hat{\mathcal{G}}$ can be initialized as an empty graph or a pre-trained chunk hierarchy graph. $\boldsymbol{M}$ retains the frequency of each chunk in the belief set $\mathbb{B}$ and $\boldsymbol{T}$ retains the transition frequencies of visual-temporally adjacent chunks sorted by temporal lags. Temporal lag is the time difference between the end of the previous chunk and start of the next chunk. The pseudocode for the Visual-Temporal HCM is shown in Algorithm 2.

At each parsing step, online HCM does the following:

1. Identifies the chunks biggest in volume that explain observation from the time point when the last chunk ended to the current time point, and store them in the set of current chunks.

2. Identifies the currently ending chunks and their adjacent previous chunks and updates their marginal and transition counts.

3. Modifies the set of chunks used to parse the sequence based on their adjacency.

4. Entries in $M$ and $T$ are subject to memory decay at the rate of $\theta$. If any entry goes below the deletion threshold $DT$, their corresponding entries in $M, T, \mathbb{B}$ and $\hat{\mathcal{G}}$ are deleted.

If two parsed visual-temporally adjacent chunks violates the independence testing criterion and they are within the proximity of each other under a padding threshold, they are grouped together into a new chunk. The constituent parts of a chunk remains in the belief set, with the count frequency subtracted by the estimation of the joint occurrence frequency.

---

**Algorithm 2:** Online HCM

---

**input** : Seq, $\hat{\mathcal{G}}, \theta, DT$
**output** : $\hat{\mathcal{G}}$
$M, T \leftarrow \hat{\mathcal{G}}.M, \hat{\mathcal{G}}.T$;
PreviousChunkBoundaryRecord $\leftarrow []$;                        /* Record Chunk Endings */
ChunkTerminationTime.setall(-1);
**while** *Sequence not over* **do**
    CurrentChunks, ChunkTerminationTime =
     IdentifyTheLatestChunks(ChunkTerminationTime);
    ObservationToExplain $\leftarrow$ refactor(Seq, ChunkTerminationTime);
    **for** *Chunk in CurrentChunks* **do**
        **for** *CandidateAdjacentChunk in PreviousChunkBoundaryRecord* **do**
            **if** *CheckAdjacency(Chunk, CandidateAdjacentChunk)* **then**
                $M, T, \mathbb{B}, \hat{\mathcal{G}} \leftarrow$ LearnChunking(Chunk, CandidateAdjacentChunk. $M, T, \mathbb{B}, \hat{\mathcal{G}}$);
            **end**
        **end**
        ChunkTerminationTime.update(CurrentChunks)
    **end**
    PreviousChunkBoundaryRecord.add(CurrentChunks);
    Forgetting($M, T, \mathbb{B}, \hat{\mathcal{G}}, \theta, DT$, PreviousChunkBoundaryRecord);
**end**

---

To process and update chunks online, HCM iterates through the visual temporal sequence, identifies chunks, marks the termination time corresponding to each visual dimension and stores them in ChunkTerminationTime. As multiple visual temporal chunks can be identified to occur simultaneously, CurrentChunks stores the identified chunks that have not reached their ending points.

Once one or more chunks are identified to be ending at a time point, they are stored inside PreviousChunkBoundaryRecord and their finishing time is updated for each visual pixel in ChunkTerminationTime. Corresponding entries in $M$ are updated. The chunks that finishes after the start of the current chunk is checked with each current chunk on whether there is a visual temporal adjacency in addition to a violation of the independence test.

If a pair of adjacent chunks $c_L$ and $c_R$ violate the independence testing criterion, they are combined into one chunk $c_L \oplus c_R$. A new entry is created in $M$ with the joint occurrence frequency for $c_L \oplus c_R$, this occurrence frequency is subtracted from the marginal record of $c_L$ and $c_R$. Additionally, other combinations that result in the same chunk will accumulate toward the count of $c_L \oplus c_R$. The constituents' transition entries are set to 0. As a new chunk, $c_L \oplus c_R$ inherits the adjacency entries of $c_R$, and the marginal frequencies for $c_L$ and $c_R$ are each subtracted by 1.

## E   Proof of Recoverability

As the belief set $\mathbb{B}$ keeps changing when one modifies the chunks in a sequence, so does the parsing length $N_\mathbb{B}$ and the probability associated with the belief set $P_{A_\mathbb{B}}$. This translates to a change of $N$ and a set of constraints on the probabilities defined on the augmented support set. We approach this problem in the following steps:

- Formulate the definition of probabilities based on $N$.
- Identify all relevant changes of $N$ before and after the chunk update.
- Translate this change of $N$ to the constraints on probability updates.

We derive the relation between the probabilities when two chunks $c_L$ and $c_R \in \mathbb{A}_d$ are concatenated together to form a new chunk $c_L \oplus c_R$ and update the alphabet to $\mathbb{A}_{d+1}$.

### E.0.1 Summary N

Going from $\mathbb{A}_d$ to $\mathbb{A}_{d+1}$, $c_L$ and $c_R$ are both chunks in $\mathbb{A}_d$ and merged together as a new chunk to augment $\mathbb{A}_d$. The chunks in $\mathbb{A}_d$ can be divided into three groups, $c_L$, $c_R$, and $\mathbb{A}_d \setminus \{c_L, c_R\}$. The relation between $N_d$ and $N_{d+1}$ is:

$$N_{d+1} = \left[ \sum_{c \in \mathbb{A}d - c_L - c_R} N_d(c) \right] + N_{d+1}(c_L) + N_{d+1}(c_R) + N_{d+1}(c_L \oplus c_R) \qquad (6)$$

Additionally, $N_{d+1}(c_L) = N_d(c_L) - N_d(c_L \rightarrow c_R)$, $N_{d+1}(c_R) = N_d(c_R) - N_d(c_L \rightarrow c_R)$. Chunking reduces the number of times sub-chunks are being parsed when sub-chunks occur right after each other by twofold.

$$N_d = \sum_{c \in \mathbb{A}_d} N_d(c) = \left[ \sum_{c \in \mathbb{A}_d - c_L - c_R} N_d(c) \right] + N_d(c_L) + N_d(c_R) \qquad (7)$$

Comparing the above two equations we arrive at

$$N_d(c_L) + N_d(c_R) = N_{d+1}(c_L) + N_{d+1}(c_R) + 2N_{d+1}(c_L \oplus c_R)$$

We know that $N_d(c_L \rightarrow c_R) = N_{d+1}(c_L \oplus c_R)$ and therefore: $N_{d+1}(c_L) + N_{d+1}(c_R) = N_d(c_L) + N_d(c_R) - 2N_d(c_L \rightarrow c_R)$. The relation between the parsing counts $N_d$ and $N_{d+1}$ when switching from the alphabet set $\mathbb{A}_d$ to $\mathbb{A}_{d+1}$ by chunking $c_L$ and $c_R$ in $\mathbb{A}_d$ together is:

$$N_{d+1} = \left[ \sum_{c \in A_d - c_L - c_R} N_d(c) \right] + N_d(c_L) + N_d(c_R) - N_d(c_L \rightarrow c_R) \qquad (8)$$

$$N_{d+1} = N_d - N_d(c_L \rightarrow c_R) \qquad (9)$$

### E.0.2 Marginal N

To proceed into formulating the joint probability given a particular belief space, we need to formulate how the count of $N(c)$ for a chunk changes when the belief space when switching from $\mathbb{A}_d$ to $\mathbb{A}_{d+1}$, with the same division as before.

Of course, the count function should be fixed. However, the probability function associated with the chunks will change based on the update of the belief set. We use the update of the count function to find the relation between the probability updates.

For all $x$ in $\mathbb{A}_d - \{c_L, c_R, c_L \oplus c_R\}$: $N_{d+1}(x) = N_d(x)$, $N_{d+1}(c_R) = N_d(c_R) - N_d(c_L \rightarrow c_R)$, $N_{d+1}(c_L) = N_d(c_L) - N_d(c_L \rightarrow c_R)$, and $N_{d+1}(c_L \oplus c_R) = N_d(c_L \rightarrow c_R)$.

### E.1 Probability Density Switch when $\mathbb{A}_d$ expands to $\mathbb{A}_{d+1}$

The constraint is: the number of counts $N$ for all chunks defined for the support set $\mathbb{A}_d$ must remain the same for the support set $\mathbb{A}_{d+1}$, so that the definition of $P_{\mathbb{A}_d}$ for all relevant chunks within $\mathbb{A}_d$ remains the same when $\mathbb{A}_d$ expands to $\mathbb{A}_{d+1}$.

The probability of a chunk occurring in the alphabet set $\mathbb{A}_d$ is defined as: $P_{\mathbb{A}_d}(c) = \lim_{N_d \rightarrow \infty} \frac{N_d(c)}{N_d}$.

Because $N_d$ and $N_{d+1}$ are only a constant away, both go to infinity if one of them does, so there is a relation between the definition of probability $P_{\mathbb{A}_d}(c)$ and $P_{\mathbb{A}_{d+1}}(c)$. For any chunk $x$ in $\mathbb{A}_d$ that is not $c_L$ and $c_R$, $N_{d+1}(x) = N_d(x)$, $P_{A_{d+1}}(x) = \lim_{N_{d+1} \rightarrow \infty} \frac{N_{d+1}(x)}{N_{d+1}} = \lim_{N_d \rightarrow \infty} \frac{N_d(x)}{N_d - N_d(c_L \rightarrow c_R)}$.

That is, the probability of a chunk of this category at d and d+1 satisfies this relationship that

$$P_{A_{d+1}}(x) = P_{A_d}(x)\frac{\lim_{N_d \to \infty} N_d}{\lim_{N_{d+1} \to \infty} N_d - N_d(c_L \to c_R)}.$$

For $c_L$ and $c_R$ in $A_{d+1}$: $P_{A_{d+1}}(c_L) = \lim_{N_{d+1} \to \infty} \frac{N_{d+1}(c_L)}{N_{d+1}}$, $P_{A_d}(c_L) = \lim_{N_{d+1} \to \infty} \frac{N_d(c_L)}{N_d}$, $P_{A_{d+1}}(c_R) = \lim_{N_{d+1} \to \infty} \frac{N_{d+1}(c_R)}{N_{d+1}}$, $P_{A_d}(c_R) = \lim_{N_d \to \infty} \frac{N_d(c_R)}{N_d}$.

Since $N_{d+1}(c_L) = N_d(c_L) \oplus N_d(c_L \oplus c_R)$, $P_{A_{d+1}}(c_L) = \lim_{N_{d+1} \to \infty} \frac{N_d(c_L) - N_d(c_L \oplus c_R)}{N_{d+1}}$, $P_{A_{d+1}}(c_L) = \lim_{N_d \to \infty} \frac{N_d(c_L) - N_d(c_L \to c_R)}{N_d - N_d(c_L \to c_R)}$, $P_{A_{d+1}}(c_R) = \lim_{N_d \to \infty} \frac{N_d(c_R) - N_d(c_L \to c_R)}{N_d - N_d(c_L \to c_R)}$

Finally, $P_{A_{d+1}}(c_L \oplus c_R) = \lim_{N_{d+1} \to \infty} \frac{N_{d+1}(c_L \oplus c_R)}{N_{d+1}}$, $P_{A_d}(c_L \oplus c_R) = \lim_{N_d \to \infty} \frac{N_d(c_L \to c_R)}{N_d}$.

Since $N_d(c_L \to c_R) = N_{d+1}(c_L \oplus c_R)$, we have $P_{A_{d+1}}(c_L \oplus c_R) = \lim_{N_d \to \infty} \frac{P_{A_d}(c_L \oplus c_R) N_d}{N_{d+1}}$.

For summary probabilities: $N_{d+1} = N_d - N_d(c_L \oplus c_R) = N_d - N_d P_d(c_L \oplus c_R)$, and $\frac{N_{d+1}}{N_d} = 1 - P_d(c_L \oplus c_R)$.

### E.1.1 Marginal Probabilities

The next level marginal probability follows the constraints when the support set changes from $A_d$ to $A_{d+1}$: $P_{d+1}(x) = \frac{P_d(x)}{1 - P_d(c_L \oplus c_R)}$, $P_{d+1}(c_R) = \frac{P_d(c_R) - P_d(c_L \oplus c_R)}{1 - P_d(c_L \oplus c_R)}$, $P_{d+1}(c_L) = \frac{P_d(c_L) - P_d(c_L \oplus c_R)}{1 - P_d(c_L \oplus c_R)}$, $P_{d+1}(c_L \oplus c_R) = \frac{P_d(c_L \oplus c_R)}{1 - P_d(c_L \oplus c_R)}$.

### E.2 Hierarchical Generative Model

At the beginning of the generative process, the atomic alphabet set $A_0$ is specified. Another parameter, $d$, specifies the number of additional chunks that are created in the process of generating the hierarchical chunks. Starting from the alphabet $A_0$ with initialized elementary chunks $c_i$ from the alphabet, the probability associated with each chunk $c_i$ in $A_0$ needs to satisfy the following criterion:

$$\sum_{c_i \in A_0} P_{A_0}(c_i) = 1 \tag{10}$$

Meanwhile, $P(c_i) \geq 0, \forall c_i \in \mathbb{A}_0$.

We assume that at each step the marginal and transitional probability of the previous steps are known. The next chunk is chosen as the combined chunks with the biggest probability. The order of construction in the generative model follows the rule that the combined chunk with the biggest probability on the support set of pre-existing chunk sets is chosen to be added to the set of chunks.

$$c_L \oplus c_R = \arg\max_{c_L, c_R \in \mathbb{A}_d \setminus \{0\}} P_{\mathbb{A}_d}(c_L \oplus c_R) \tag{11}$$

Under the constraint that:

$$P_{\mathbb{A}_d}(c_L) P_{\mathbb{A}_d}(c_R) \leq P_{\mathbb{A}_d}(c_L \oplus c_R) \leq \min\{P_{\mathbb{A}_d}(c_L), P_{\mathbb{A}_d}(c_R)\} \tag{12}$$

This can be calculated from the transitional and marginal probability of the previous step.

$$c_L \oplus c_R = \arg\max_{c_L, c_R \in \mathbb{A}_d \setminus \{0\}} P_{A_d}(c_L \oplus c_R) = \arg\max_{c_L, c_R \in \mathbb{A}_d \setminus \{0\}} P_{A_d}(c_L) P_{A_d}(c_R | c_L) \tag{13}$$

In practice, after the chunks are specified in $\mathbb{A}_d$, the probability value associated with chunks in $\mathbb{A}_0$ are sampled from a flat Dirichlet distribution, which is then sorted so that the smaller sized chunks contain more of the probability mass and the null-chunk carries the biggest probability mass. Then, the above constraint is checked for the assigned probability on each of the newly generated chunk with their associated alphabet set $\mathbb{A}_i$. This process repeats until the probability drawn satisfies the condition for every newly created chunk.

At first, the set of chunks are $\mathbb{A}_0$, which is assigned each as an integer. Then $d$ additional recombination processes are carried out. In each process, two chunks are randomly chosen from the pre-existent alphabet set to recombine into a new chunk, until $d$ additional chunks are being created to augment the set of chunks from $\mathbb{A}_0$ to $\mathbb{A}_d$. The Dirichlet distribution is randomly generated in an unsorted fashion, and then the biggest probability mass is assigned to 0. Constraints are checked recursively.

**Theorem 1** (Marginal Probability Space Conservation). *After the addition of $c_{d,i} \oplus c_{d,j}$ and the change of probability, $P_{\mathbb{A}_d}$ is still a valid probability distribution.*

**Proof:**

$$
\begin{aligned}
\sum_{c_{d,k} \in \mathbb{A}_d} P_{\mathbb{A}_d}(c_{d,k}) &= \sum_{c_{d,k} \in \mathbb{A}_{d-1} - c_{d-1,i} - c_{d-1,j}} P_{\mathbb{A}_{d-1}}(c_{d-1,k}) + \\
&\quad + P_{\mathbb{A}_{d-1}}(c_{d-1,i}) - P_{\mathbb{A}_{d-1}}(c_{d-1,j}|c_{d-1,i})P_{\mathbb{A}_{d-1}}(c_{d-1,i}) \\
&\quad + P_{\mathbb{A}_{d-1}}(c_{d-1,j}) + P_{\mathbb{A}_{d-1}}(c_{d-1,j}|c_{d-1,i})P_{\mathbb{A}_{d-1}}(c_{d-1,i}) \\
&= 1
\end{aligned}
\tag{14}
$$

$\square$

**Theorem 2** (Measure Space Preservation). *Given that at the end of the generative process with depth $d$ one ends up having an alphabet set $\mathbb{A}_d$, the probability space defined on $\mathbb{A}_i$, which includes the marginal and joint probability of any chunk and combinations of chunks in $\mathbb{A}_i$, $i = 0, 1, 2, \ldots d$, which are predecessor alphabet sets of $\mathbb{A}_d$, all values in the set $\mathbb{M}_d$ and $\mathbb{T}_d$ remain the same no matter how the future support set changes according to the generative model.*

**Proof:** By induction.

- Base case: starting from the initialized alphabet set $\mathbb{A}_0$, the probability of $P_{A_0}(c), c \in \mathbb{A}_0$, and the probability of $P_{A_1}(xy), x, y \in \mathbb{A}_0$, for all valid c, x, y, when the alphabet is $\mathbb{A}_1$. Going from $\mathbb{A}_0$ to $\mathbb{A}_1$, $N_0(c)$, $N_0$ and $N_0(x \to y)$ does not change, therefore $P_{A_0}(c)$ and $P_{A_0}(x \to y)$ at the alphabet $\mathbb{A}_1$ is the same as that when the alphabet is $\mathbb{A}_0$.

- Induction Step: starting from the initialized alphabet set $\mathbb{A}_d$, the probability of $P_{A_d}(c), c \in \mathbb{A}_d$, and the probability of $P_{A_d}(xy), x, y \in \mathbb{A}_d$, for all valid c, x, y, when the alphabet is $\mathbb{A}_{d+1}$. Going from $\mathbb{A}_d$ to $\mathbb{A}_{d+1}$, $N_d(c)$, $N_d$ and $N_d(x \to y)$ does not change, therefore $P_{A_d}(c)$ and $P_{A_d}(x \to y)$ at the alphabet $\mathbb{A}_{d+1}$ is the same as that when the alphabet is $\mathbb{A}d$.

$\square$

**Theorem 3.** *The order of $P_{A_i}(xy), x, y \in A_i$ for any $i = 0, 1, 2, \ldots d$ at any previous belief space is preserved throughout the update.*

**Proof:** At the end of the generative process with depth $d$, one ends up having such an alphabet set: $\mathbb{A}_d$. The probability space defined on $\mathbb{A}_i$, which includes the marginal and joint probability of any chunk and combinations of chunks in $\mathbb{A}_i$, $i = 0, 1, 2, \ldots d$ is preserved, hence the order is preserved.

$\square$

The generative process can be described by a graph update path. The specification of the initial set of atomic chunks $\mathbb{A}_0$ corresponds to an initial graph $\mathcal{G}_0$ with the atomic chunks as its vertices. At the i-th iteration, as the generative graph goes from $\mathcal{G}_{\mathbb{A}_i}$ to graph $\mathcal{G}_{\mathbb{A}_{i+1}}$, two none zero chunks $c_L, c_R$ chosen from the pre-existent set of chunks $\mathbb{A}_i$ and are concatenated into a new chunk $c_L \oplus c_R$, augmenting $\mathbb{A}_i$ by one to $\mathbb{A}_{i+1}$. The vertex set also increments from $V_{\mathbb{A}_i}$ to $V_{\mathbb{A}_{i+1}} = V_{\mathbb{A}_i} \cup c_L \oplus c_R$. Moreover, two directed edges connecting the parental chunks to the newly-created chunk are added to the set of edges: $E_{\mathbb{A}_i}$ to $E_{\mathbb{A}_i} = E_{\mathbb{A}_i} \cup (c_L, c_L \oplus c_R) \cup (c_R, c_L \oplus c_R)$. The series of graphs created during the chunk construction process going from $\mathcal{G}_{\mathbb{A}_0}$ to the final graph $\mathcal{G}_{\mathbb{A}_d}$ with $d$ constructed chunks can be denoted as a graph generating path $P(\mathcal{G}_{\mathbb{A}_0}, \mathcal{G}_{\mathbb{A}_d}) = (\mathcal{G}_{\mathbb{A}_0}, \mathcal{G}_{\mathbb{A}_1}, \mathcal{G}_{\mathbb{A}_2}, ..., \mathcal{G}_{\mathbb{A}_d})$.

### E.3 Learning the Hierarchy

The rational chunking model is initialized with one minimally complete belief set, the learning algorithm ranks the joint probability of every possible new chunk concatenated by its pre-existing belief set, and picks the one with the maximal occurrence joint probability on the basis of the current set of chunks as the next new chunk to enlarge the belief set. With the one-step agglomerated belief set, the learning model parses the sequence again. This process repeats until the chunks in the belief set pass the independence testing criterion.

**Theorem 4** (Learning Guarantees on the Hierarchical Generative Model). *As $N \to \infty$, the chunk construction graph learned by the model $\hat{\mathcal{G}}$ is the same as the chunk construction graph of the generative model: $\hat{\mathcal{G}} = \mathcal{G}$, which entails that they have the same vertex set: $\hat{\mathbf{V}} = \mathbf{V}_\mathcal{G}$ and the same*

*edge set:* $\hat{\mathbf{E}} = \mathbf{E}_{\mathcal{G}}$. *Additionally, the belief set learned by the chunk learning model* $\mathbb{B}_d = \mathbb{A}_d$, *and the marginal probability evaluated on the learned belief set* $M_{\mathbb{B}_d}$ *associated with each chunk is the same as the marginal probability imposed by the generative model on the generative belief set* $M_{\mathbb{A}_d}$.

**Proof:** Given that all of the empirical estimates are the same as the true probabilities defined by the generative model, we prove that starting with $\mathbb{B}_0$, the learning algorithm will learn $\mathbb{B}_D = \mathbb{A}_D$. $\mathbb{A}_D$ is the belief set imposed by the generative model. We approach this proof by induction.

**Base Step:** As the chunk learner acquires a minimal set of atomic chunks that can be used to explain the sequence at first, the set of elementary atomic chunks learned by the model is the same as the elementary alphabet imposed by the generative model, i.e. $\mathbb{B}_0 = \mathbb{A}_0$. Hence, the root of the graph, which contains the nodes without their parents, is the same, $\hat{\mathcal{G}} = \mathcal{G}$; put differently, $\hat{\mathbf{V}}_0 = \mathbf{V}_0$

Additionally, the learning model approximates the probability of a specific atomic chunk as $\hat{P}_{\mathbb{A}_0}(a_i)$. As $n \to \infty$, for all chunks $c$ in the set of atomic elementary chunks in $\mathbb{B}_0$, the empirical probability evaluated on the support set is the same as the true probability assigned in the generative model with the alphabet set $\mathbb{A}_0$:

$$\hat{P}_{\mathbb{B}_0}(c) = \lim_{n \to \infty} \frac{N_0(c)}{N_0} = P_{\mathbb{A}_0}(c) \tag{15}$$

**Induction hypothesis**: Assume that the learned belief set $\mathbb{B}_d$ at step $d$ contains the same chunks as the alphabet set $\mathbb{A}_d$ in the generative model.

The HCM, by keeping track of the transition probability between any pairs of chunks, calculates $\hat{P}_{\mathbb{B}_d}(c_i|c_j)$ for all $c_i, c_j$ in $\mathbb{B}_d$. Afterwards, it finds the pair of chunks $c_i, c_j$, such that the chunk created by combining $c_i$ and $c_j$ together contains the maximum joint probability violating the independence test as candidate chunks to be combined together.

$$\hat{P}_{\mathbb{B}_d}(c_i \oplus c_j) = \sup_{c_i, c_j \in \mathbb{B}_d} \hat{P}_{\mathbb{B}_d}(c_i)\hat{P}_{\mathbb{B}_d}(c_j|c_i) \tag{16}$$

We know that in the generative step the suprimum of the joint probability with the support set $\mathbb{A}_d$ is being picked to form the next chunk in the representation graph, so each step of the process at step $d$ satisfies the condition that:

$$P_{\mathbb{A}_d}(c_i \oplus c_j) = \sup_{c_i, c_j \in \mathbb{A}_d} P_{\mathbb{A}_d}(c_i)P_{\mathbb{A}_d}(c_j|c_i) \tag{17}$$

Since $P_{\mathbb{A}_d}(c_j|c_i) = \hat{P}_{\mathbb{B}_d}(c_j|c_i)$, $P_{\mathbb{A}_d}(c_i) = \hat{P}_{\mathbb{B}_d}(c_i)$, the chunks $c_i$ and $c_j$ chosen by the learning model will be the same ones as those created in the generative model.

**End step**: The chunk learning process stops once an independence test has been passed, which means that the sequence is better explained by the current set of chunks than any of the other possible next-step chunk combinations. This is the case once the chunk learning algorithm has learned a belief set $\mathbb{B}_d$ that is the same as the generative alphabet set $\mathbb{A}_d$. At this point $\hat{\mathcal{G}} = \mathcal{G}$ □

# F Experiment Detail

## F.1 Chunk Recovery and Convergence

To test the model's learning behavior on this type of sequential data, random graphs of chunk hierarchies with an associated occurrence probability for each chunk are specified by the hierarchical generative process. To do so, an initial set of specified atomic chunks $\mathbb{A}_0$ and a pre-specified level of depth (new chunks) $d$ is used to initiate the generation of a random hierarchical generative graph $\mathcal{G}$. In total, there are $|\mathbb{A}_0| + d$ number of chunks in the generative alphabet $\mathbb{A}$, with chunk $c$ having an occurrence probability of $P_{\mathbb{A}}(c)$ on the sample space $\mathbb{A}$. Once a hierarchical generative model is specified, it is then used to produce training sequences with varying length $N$ to test the chunk recovery.

To compare representation learned by the rational chunking model with the ground truth generative model $\mathcal{G}$, a discrete version of Kullback–Leibler divergence is used to evaluate learning performance:

$$KL(P||Q) = \sum_{c \in \mathbb{A}} P_{\mathbb{A}}(c) \log_2\left(\frac{P_{\mathbb{A}}(c)}{Q_{\mathbb{A}}(c)}\right) \tag{18}$$

$P_\mathbb{A}(c)$ is defined by the generative model. $Q(c)$ is the learned probability of chunk $c$. To evaluate $Q_\mathbb{A}(c)$, the learned representation is used to produce "imagined" sequences of length 1000. After that, the occurrence probability of each chunk $c$ in $\mathbb{A}(c)$ is used to evaluate $Q$, comparing the HCM's learned representation with the ground truth.

For comparison, we used the same sequence used for training HCM to train a 3-layer recurrent neural network (one embedding layer with 40 hidden units, one LSTM layer with drop-out rate = 0.2, and one fully connected layer, batch size = 5, sequence length = 3, epoch = 1, so that the data used for training is the same as N) of the training sequence, and used it to generate predictive sequences with of length 1000. The predicted sequence is parsed in the unit of the generative alphabet, producing a discrete distribution on the same support set of the generative model. This distribution is used to calculate KL. The rational chunking model is trained on the the sequence $S$ with increasing sizes (from 100 to 3000 with steps of 100) produced by the hierarchical generative graph. For each depth $d$, 5 random chunk hierarchies with the same depth is randomly assigned. The sequence generated by these random chunking graphs are then used to train both HCM and RNN.

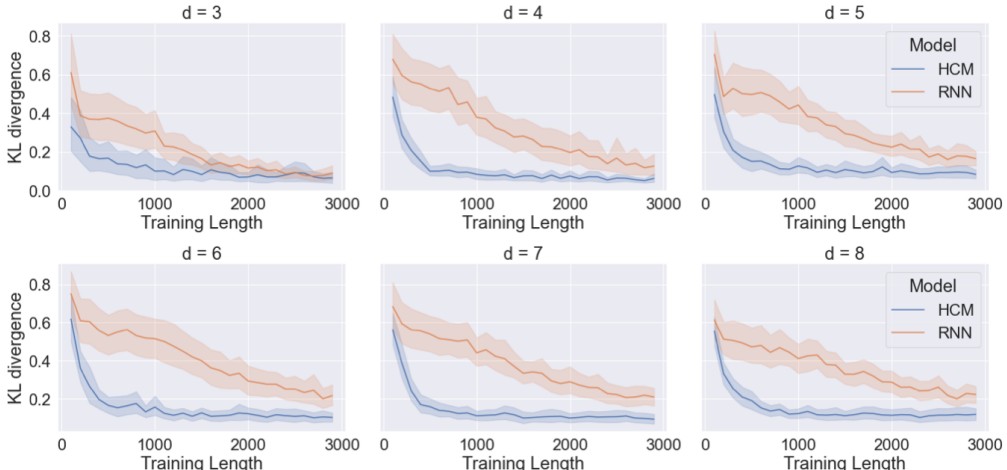

Figure 1: Learning Comparison Between HCM and RNN with Varying Graph Depth

## F.2 Transfer Between Environments with Overlapping and Interfering Structure

After training on a sequence, HCM acquires an interpretable representation. Knowing what the model has learned enables us to directly know what type of hierarchical environment would facilitate or interfere with the learned representations.

More formally, two HCM models might have acquired different hierarchical chunking graphs $\mathcal{G}_i$ and $\mathcal{G}_j$ from their past experience. These might lie on the graph construction path $(\mathcal{G}_0, \mathcal{G}_1, \mathcal{G}_2, ..., \mathcal{G}_d)$. The HCM with a chunk hierarchy graph 'closer' to the ground truth chunk $\mathcal{G}_d$ on the path, takes fewer iteration to arrive at $\mathcal{G}_d$. This also applies when the chunk hierarchies starting out are not along the graph construction path but only showing partial overlap. In other words, if $D(\mathcal{G}_i, \mathcal{G}_d) \leq D(\mathcal{G}_j, \mathcal{G}_d)$ then representation learned by HMC with graph structure $\mathcal{G}_i$ is more facilitative than that with graph structure $\mathcal{G}_j$.

Similarly, the chunk hierarchy $\mathcal{G}_i$ learned by an HCM might facilitate its performance in a new environment where $\mathcal{G}_i$ lies along the graph construction path to the true $\mathcal{G}_d$, i.e. there is partial overlap between the chunk hierarchies.

We took a graph with the trained representation and make it learn from sequences generated from a transfer and interfering environment. In the mean time, we used naive models separately to learn representations from the facilitative and interfering environment with an increasing sequence length from 50 to 1000. For all of the training models, the forgetting rate is set to $0.996$, with the

deletion threshold being $0.01$. An imaginative sequence with length $1000$ is used to evaluate the discrepancies between the non-naive learner, and the naive learner, with respect to the corresponding facilitative/interfering environment.

Note that with the model that started out from a representation learned in an interfering environment, it learns representation from the new environment as shown in figure 2, albeit slower than the naive model.

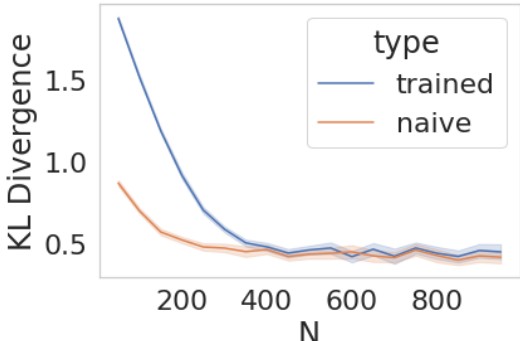

Figure 2: The model started out from a representation learned in an interfering environment converges in learning eventually, albeit slower than the naive model.

## F.3 Visual Hierarchical Chunks

The visual hierarchical chunks are crafted as binary arrays. The dark pixels are encoded as 1 and the background 0. Each image in the generative hierarchy is 25 dimensional (5 x 5) in the visual domain and size 1 in the temporal domain. An empty array is included to denote no observation. The alphabet $\mathbb{A}$ of the generative model include all 14 images in the generative hierarchy. In the generative hierarchy, higher level chunks are a composition of the lower level chunks. The occurrence probability for each generative visual chunk is drawn from a flat Dirichlet distribution with the empty observation retaining the highest mass, to emulate the sparsity of observation signals.

$$f(x_1, ..., x_k; \alpha_1, ..., \alpha_K) = \frac{1}{\mathbf{B(a)}} \prod_{i=1}^{K} x_i^{\alpha_i - 1} \tag{19}$$

Where the beta function when expressed using gamma function is: $\mathbf{B(a)} = \frac{\prod_{i=1}^{K} \Gamma(\alpha_i)}{\Gamma(\sum_i^K \alpha_i)}$, and $\mathbf{a} = (\alpha_1, .., \alpha_K)$. The parameters $(\alpha_1, .., \alpha_K)$ with $K = |\mathbb{A}|$ are all set to one.

To generate the training sequence, $P_{\mathbb{A}}(c)$, $c_1, ..., c_K \in \mathbb{A}$ is assigned by the sampled distribution. Each image $c$ in the hierarchy are sampled independently with probability $P_{\mathbb{A}}(c)$ and appended to the end of the sequence. As a result, there are visual correlations in the sequence defined by the hierarchy, but temporally, each image slice is sampled independently. In total, the sequence is made up of 2000 images.

We use online HCM to learn representation from visual-temporal sequences (forgetting rate = 0.996, deletion threshold = 0.01). HCM learn to construct representations from simple to complex (representation snapshots are collect at $t = 10$, $t = 100$, and $t = 1000$ respectively). Figure 3 shows the construction images from simple to complex. Parent chunks are made from the concatenation of their children chunks.

## F.4 GIF Movement

Here, we take a GIF file of a squid jumping in the sea with bubbles rising in the background. The entire animation is made up of 10 frames of $25 \times 25$ images. Each unique color of the GIF is mapped to an integer, with the background having a value of 0. In this way, the animation sequence

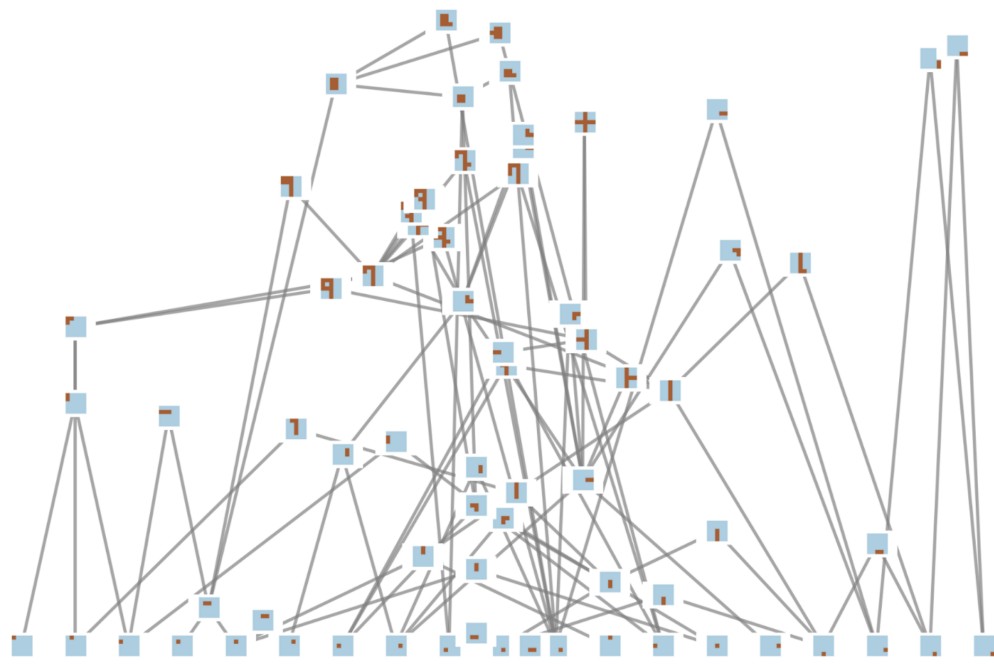

Figure 3: Construction of Hierarchy for Visual Chunks

becomes an integer array with size $T \times H \times W$. $T$ is the temporal dimension, $H$ and $W$ are the respective spatial dimension of the sequence. In this way, the gif file is converted into a tensor with size $10 \times 25 \times 25$. The entire movement is repeated 100 times and trained on the online version of HCM (forgetting rate = 0.996, deletion threshold = 0.01). Images are taking from the chunk learning graph of HCM at the end of the training process.

### F.5 Human Experiment

In the dataset from Wu et al. (2022), 47 participants are recruited from Amazon Mechanical Turk for a sequence learning experiment. Specifically, they conduct a serial-reaction-time task. In this task, participants are instructed to press the corresponding key on the keyboard upon observation of consecutive sequential instructions. Participants are rewarded based on a combination of speed and accuracy. Particularly, the training sequence is made up of sampling from the chunk [1,2,3] and [4] independently without pauses in between. Participants chunking behavior inferred from the reaction-time speed up is provided. If participants would be confident that some particular sequential instructions will show up, then they will be more confident to predict within-chunk items compare to between-chunk items and thereby speed up their reaction time.

The average chunk size across the training sequence is evaluated by averaging the size with a window of 30 chunks. Longer chunks imply that the predictive horizon, i.e. how confident participants can predict the upcoming sequential instructions, increases with practice.

To evaluate chunk learning of RNN on the same sequence, the probability estimate of each instruction choice in RNN is compared with the human data by evaluating the predicted negative log probability of the upcoming sequential instruction as a proxy of reaction time, since reaction time is often modeled as the negative log of choice probability. This reaction time is then grouped into within and between-chunk reaction time using mixtures of Gaussian classification method as in Wu et al. (2022).

We run online HCM on the same training sequence (forgetting rate = 0.90, deletion threshold = 0.1). We recorded the sequence of chunks in addition to the probability of chunk activation as calculated

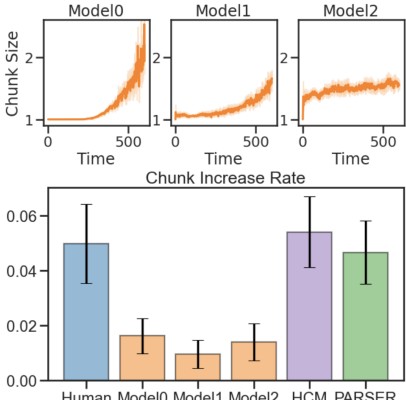

Figure 4: Comparison of chunk increase rate across three RNN models. Model0 is the architecture used in the main paper.

from the marginal frequencies. The average chunk size evaluated on the chunk sequence is used as a measure to compare with humans and RNNs. Additionally, the probability of within-chunk reaction time is set to $1 - 4\epsilon$ with $\epsilon = 0.05$ denoting the probability of choosing instructions outside of the predicted chunks.

### F.5.1 Size of RNN

We compared the chunk size increase rate across three RNN models varying in size as in Figure 4. Model0 is the RNN that we used in the experiment, which has the dimension of 40 embedding dimensions, and 3 layers, each with 40 LSTM units and a dropout rate of 0.2, followed by a fully connected feed-forward layer.

Model1 reduces the size of Model0 by half. Model1 has 20 embedding dimensions, and 3 layers, each with 20 LSTM units and the same dropout rate, followed by a fully connected feed-forward layer.

Model2 is about two times the size of Model0, with the same 40 embedding dimensions, and 5 layers LSTM neurons, each layer has 40 hidden units and a dropout rate of 0.2, followed by a fully connected feed-forward layer.

In short, Model1 is half of the size of Model0, and Model2 is double the size of Model0. Across all three RNN architectures, the chunk size only increased very slowly with increasing sequence length.

### F.6 fMRI dataset

The data comes from the brain development dataset (fMRI), including the measurement of 50 children (ages 3 - 13) and 33 young adults (ages 18 - 39). The experiment measures the resting state activities of subjects in the scanner, watching the PIXAR movie 'Partly Cloudy'. The data is down sampled to 4mm resolution, with a repetition time (TR) of 2 secs. Each session translates to 168 TRs in total. Signals of the fMRI BOLD activity are extracted using the MSDL labeled atlas of brain spontaneous activity Varoquaux et al. (2011) that segments regions of the brain and defines a functional parcellation of the brain's localized regions. In this way, the brain activities of each participant are extracted into a time series with 39 non-overlapping functional dimensions. Confounds from the original data file is extracted from the signal. The preprocessing pipeline, including the transformation from 4D images to 2D masked array, is obtained using the nilearn package offered by Abraham et al. (2014) heavily based on scikit-learn Pedregosa et al. (2011).

Upon exposure to a time series, online HCM (forgetting rate = 1.0, deletion threshold = 0.1) constructs chunks from their constituents and arrives at a nested hierarchy of chunk relations. The independent chunk activation frequencies $n$ within the hierarchy are identified upon another independent parse of the sequence.

In the movie, 19 scenes are tagged with their corresponding content. After running HCM on the data, one can obtain the chunk activation information after every tagged scene for each participant.

The 19 scenes are then categorized into 5 coarse categories: anger, pain, social, compassion, and sadness. The chunk activation probability is evaluated as the probability of one chunk being identified within 3 TRs after a tagged scene happens. Thereby, one can arrive at the chunk activation probability for each subject with the five tagged emotional categories.

### F.6.1 More examples

Figure 5 shows more examples of scene-tagged brain chunk activities.

An additional example of hierarchical structure learned by HCM that describes the nested hierarchical relationship between brain activation regions that are related to social and emotion recognition as well as theory-of-mind circuits is shown in Figure 6.

Figure 7 shows the conditional probability of scene contents given the activation of size-2 chunks. Brain functional regions such as 'D ACC' and 'R A Ins' show a multi-faceted prospect in reacting to scenes with a diverse range of categories, whereas chunks such as 'Striate' and 'R Par' activate only to the scene when it contained greetings and a hug.

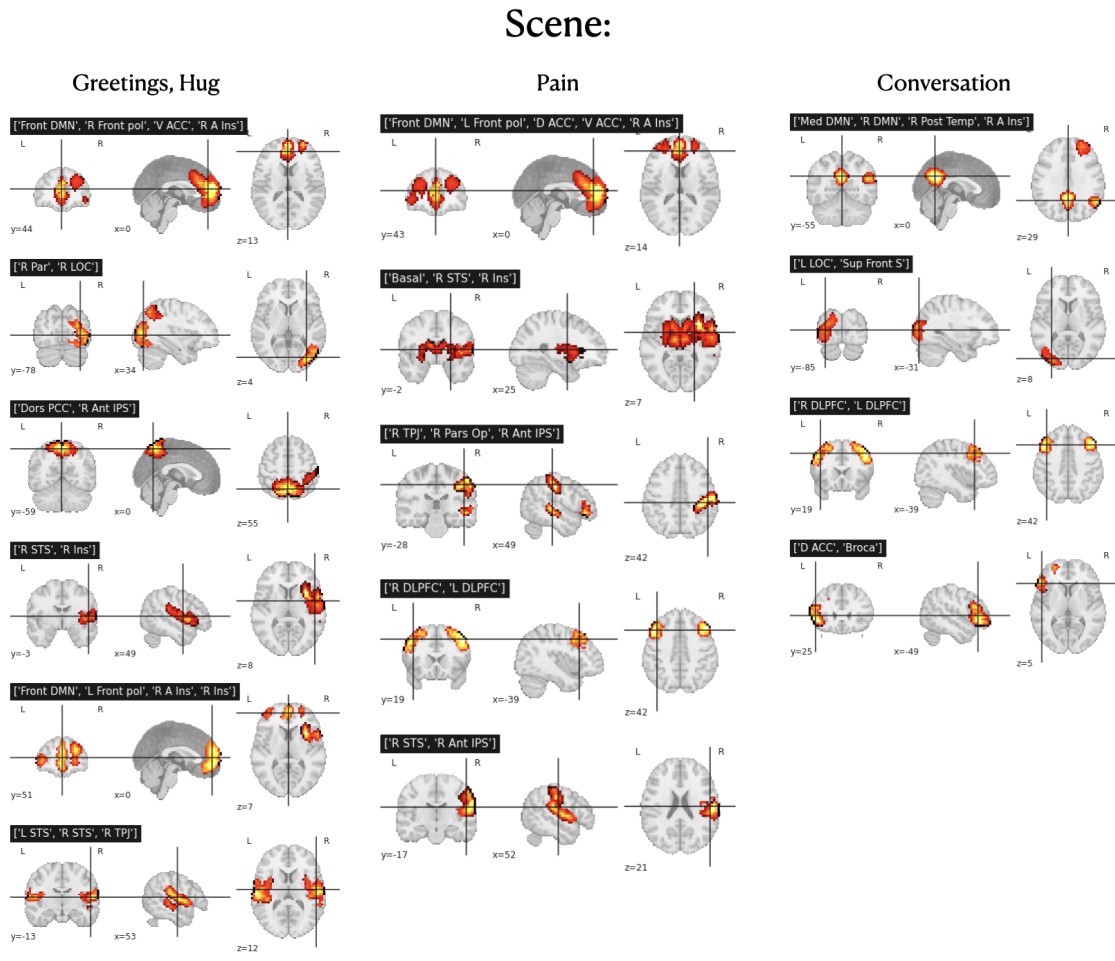

Figure 5: Additional Examples of Scene-Tagged Chunks of Brain Area Activation.

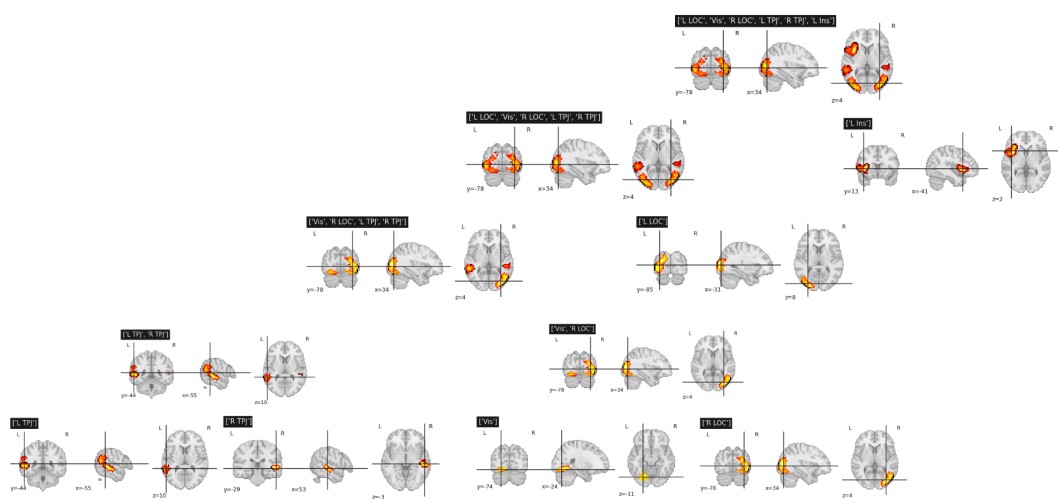

Figure 6: Additional example of hierarchical relationship between activation chunks of brain areas.

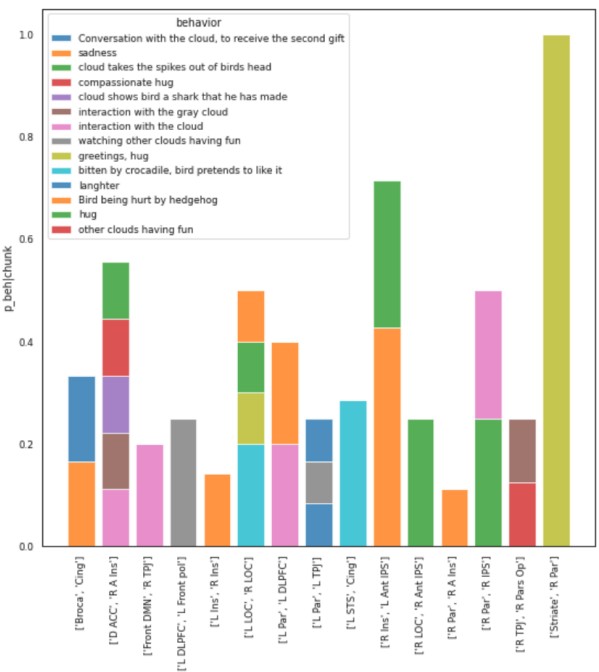

Figure 7: Conditional probability chunk activation given scene content.

# G   Translating representation learned by HCM into $n$th order Markov Chain

Given an HCM that has learned a set of chunks $\mathbb{B}$. The storage of this representation demands the storage of $|\mathbb{B}|$ number of frequencies for each learned chunk entry, and the storage of the transition probability in a matrix with size $|\mathbb{B}| \times |\mathbb{B}|$.

Translating this HCM representation into an $n$th order Markov chain in the most parsimonious way would demand the storage of each individual state specific to each chunk. Thereby, a Markov chain

that contains the full information as in the chunks and transitions between chunks learned by HCM requests the number of states $n = \sum_{c \in \mathbb{B}} |c|$, where $|c|$ denotes the size of each chunk. So this Markov chain needs to store the probability of $\sum_{c \in \mathbb{B}} |c|$ number of states and a transition probability matrix of $\sum_{c \in \mathbb{B}} |c| \times \sum_{c \in \mathbb{B}} |c|$, which is a matrix much bigger than $|\mathbb{B}| \times |\mathbb{B}|$.

With regard to chunk size, for a learned graph with hierarchy depth $d$, the worst case scenario of the maximum chunk size will be $2^d$. The upper bound of the maximum chunk size grows exponentially with depth $d$. Therefore, the number of states for an equivalent Markov chain will be bounded by $|\mathbb{B}| \times 2^d$.

## H   Memory Analysis

We analyze the memory demand to store each chunk in the case of optimal encoding, and provide examples of why chunking is beneficial for encoding by formulating an Expected Unit of Explanatory Power (EUEP) measure.

Given a set of chunks $\mathbb{B}$, each chunk $c \in \mathbb{B}$ with size $|c|$. The observational sequence $S$ is tiled by chunks. The minimal code length $I(c_i)$ assigned to each chunk $c_i \in \mathbb{B}$ to distinguish one from another is bounded by the optimal code length $- \log_2 P(c_i)$.

$$I(c_i) = - \log_2 P(c_i) \tag{20}$$

A chunk that occurs quite frequently contains a low information content, and correspond to a small code length, hence it is easier for the agent to encode this chunk.

The average length of sequence spent per bit of information to store a specific chunk $c_i$ is: $\frac{|c_i|}{I(c_i)}$.

The bigger this length per bit ratio is, the more efficient storage is optimized to encode a chunk $c_i$ with occurrence probability $P(c_i)$.

We denote the expectation of this unit length per code across all possible chunks as **Expected Unit Explanatory Power** (EUEP):

$$EUEP = \sum_{c_i \in \mathbb{B}} p(c_i) \frac{|c_i|}{I(c_i)} = \sum_{c_i \in \mathbb{B}} p(c_i) \frac{|c_i|}{- \log_2 P(c_i)} \tag{21}$$

Note $I(c_i)$ denotes the information content for a specific event.

### H.1   Example

Given the following sequence $S$:
$$11122222111 2222$$
We compare two belief sets $\mathbb{B}_1 = \{111, 2222\}$ with $c_1$ being 111 and $c_2$ being 2222 and the second belief set $\mathbb{B}_2 = \{1, 2\}$ with $c_1$ and $c_2$ being the single atomic units in the sequence.

In the first case, $S$ will be parsed by $\mathbb{B}_1$ as $c_1, c_2, c_1, c_2$, whereas in the second case, the parsing by $\mathbb{B}_2$ will become $c_1, c_1, c_1, c_2, c_2, c_2, c_2, c_1, c_1, c_1, c_2, c_2, c_2, c_2$.

For the first example: $P(c_1) = \frac{1}{2}, P(c_2) = \frac{1}{2}$. Their corresponding information contents are:

$$I(c_1) = - \log_2(P(c_1)) = - \log_2(0.5) \tag{22}$$

$$I(c_2) = - \log_2(P(c_2)) = - \log_2(0.5) \tag{23}$$

The expectation of unit length explaining this sequence per bit are:

$$EUEP = \sum_{c_i \in \mathbb{B}_1} p(c_i) \frac{|c_i|}{I(c_i)} = 0.5 \times \frac{|111|}{- \log_2(0.5)} + 0.5 \times \frac{|2222|}{- \log_2(0.5)} = 3.5 \tag{24}$$

For the second example:

The information content for each chunk is:

$$I(c_1) = -\log_2(P(c_1)) = -\log_2(6/14) \tag{25}$$

$$I(c_2) = -\log_2(P(c_2)) = -\log_2(8/14) \tag{26}$$

The expectation of unit length explaining this sequence per bit are:

$$EUEP = \sum_{c_i \in \mathbb{B}_2} p(c_i) \frac{|c_i|}{I(c_i)} = 6/14 \times \frac{|1|}{-\log_2(6/14)} + 8/14 \times \frac{|2|}{-\log_2(8/14)} = 1.05 \tag{27}$$

From the above two examples, the first chunking method $B_1$ with chunks $\{c_1 = 111, c_2 = 2222\}$ is more efficient compared to the second chunking method $B_2$ with chunks $\{c_1 = 1, c_2 = 2\}$. It explains 3.5 sequential units per bit of encoding compared to 1.05 sequential units per bit.

# I Sensitivity Analysis

## I.1 Sensitivity to Noisy Observations

One interesting question is how sensitive is HCM's learning performance to noisy observations. To examine HCM's sensitivity to noisy observations, we generate sequences from random hierarchical generative graphs with an increasing level of depth (from 3 to 8), taking 5 sample graphs of each level of depth. Each of the random graphs is used to generate training sequences increasing from length 100 to 900.

To simulate noisy observations, we add an $\epsilon$ probability of switching to an alternative atomic unit for each unit in the sequence. That is, for each sequential element, there is a $(1 - \epsilon)$ probability that the element stays the same, and a $\epsilon$ probability of flipping to any other atomic unit with equal probability. We took an exponentially increasing level of $\epsilon$ ranging from 0 to 0.1, and trained the rational HCM algorithm on these noise-perturbed sequences.

Shown in Figure 8 is the learning performance with an increasing depth of the random generative model. The learned representation is evaluated on the underlying ground truth in the generative model. Learning converges in all cases. Thus, HCM's performance is fairly robust to low levels of noise.

## I.2 Sensitivity to Phase Shift

Another question we investigated was the performance of the model in response to a phase shift of the sequence, in other words, where the start of the sequence is. To investigate this question, we generated random hierarchical chunking model with depth $d = 5$ of random hierarchical graphs. For each generative model, we shifted the sequence rightwards $n$ steps, ranging from 0 to 19, and evaluated the learning performance with an increasing length of the sequence. In Figure 9, we plot the sensitivity of of HCM's learning convergence to phase shift. There was no systematic influence of phase on learning performance.

# J Comparison with PARSER

To position HCM better in relation to other models, we compared HCM's fitting on human data with the most related cognitive algorithm, PARSER.

Figure 10 shows the comparison of chunk size increase between HCM, RNN, PARSER and human behavior. HCM and PARSER continue to learn and build up longer chunks as they go through the sequence. Evaluating the average rate of chunk growth also showed that both PARSER and HCM are more similar to participants' than the RNNs. The negative log-probabilities of sequence elements generated by the HCM, RNN and PARSER were both significantly related to human reaction times (that reflect the certainty of their internal predictions). Yet the relationship was substantially stronger between HCM ($\beta = 16.74$, $p \leq 0.001$) and human participants than PARSER ($\beta = 11.070$, $p \leq 0.001$) compared to that of the RNN ($\beta = 9.24$, $p \leq 0.001$).

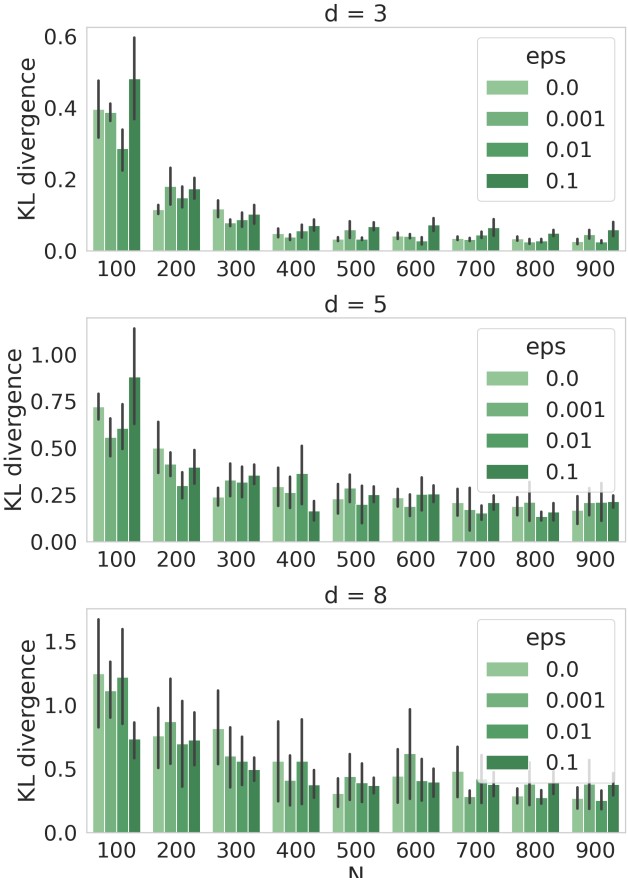

Figure 8: Learning Performance of HCM with an increasing level of noise-perturbed sequence.

## K   Learning Motifs

There are situations when there is an underlying structure governing seemingly disparate sequences, for example, the sequences "12221212" and "34443434". We show a method to use HCM to learn such underlying structure. We denote such structures as motifs, and formulate a projecting function that maps a sequence from the observational space to some projected space, followed by illustrations of examples showing how the motifs can be learned by HCM. Finally, we show an experiment demonstrating HCM's motif learning ability.

**Definition 12 (*Projecting Function $f(S)$*)**

*A projecting function $f : \mathbb{O} \to \mathbb{P}$ maps a sequence in observation space to a projected space.*

The projection almost always maps from a higher dimensional space to a lower dimensional space because the intention is to discover common, overlapping parts that are shared between disparate sequences in the observation space.

**Definition 13 (*Motif*)**

*A motif $m$ is a chunk in the projected space, made up of concatenation of elements in the projected space.*

**Definition 14 (*Injection Function $g(m) \to c$*)**

*An injection function maps from motifs back to chunks in observational space $f : \mathbb{P} \to \mathbb{O}$.*

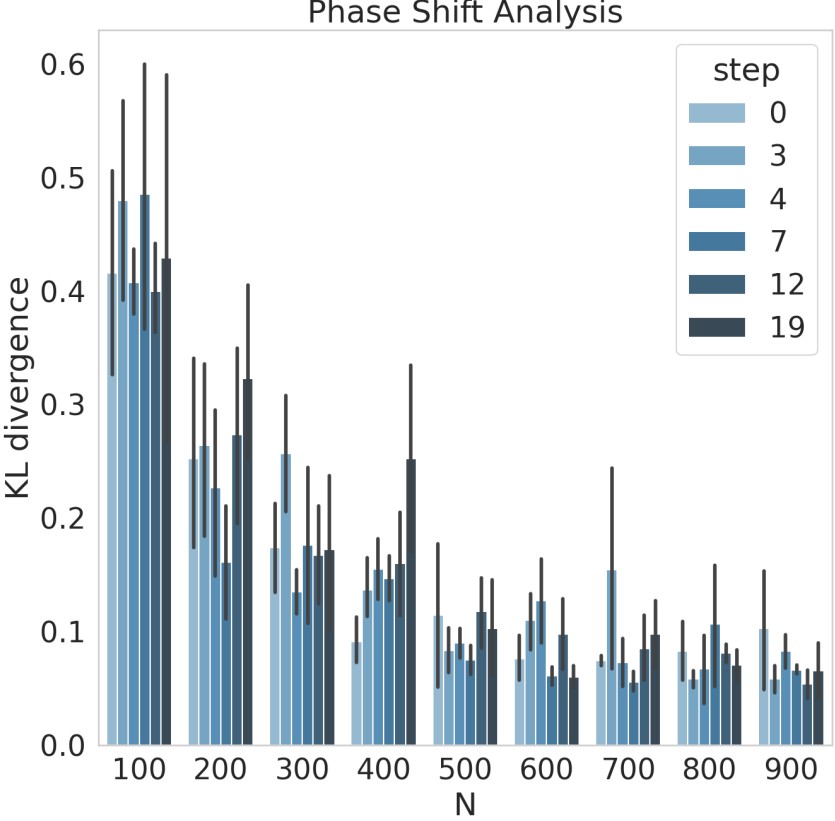

Figure 9: Learning performance of HCM with increasing phase shift steps.

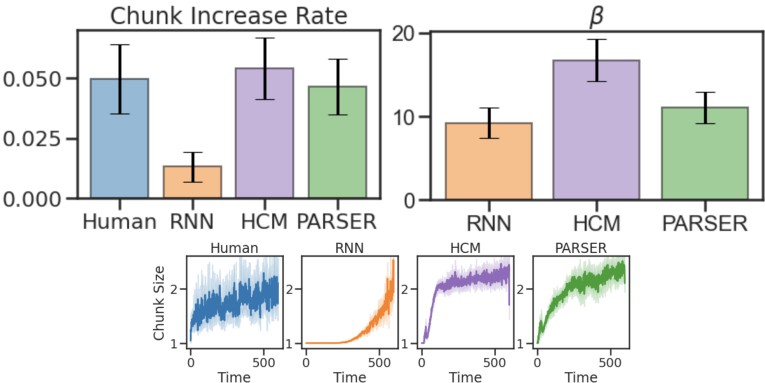

Figure 10: Model comparison for human SRT data.

## K.1 Example

We here demonstrate an example of observational sequences with any underlying motif, and illustrate why motif learning is useful for encoding or prediction. The projecting function, when applying to sequences with an underlying motif, is used so that similar motifs between projected representations can be clustered and identified as a whole.

Let's say the model has observed the following sequences: $ABBAAABBB$, $CDDCCCDDD$, and $EFFEEEFFF$: The observation set is $\mathbb{O} = \{A, B, C, D, E, F\}$, which are the set of observations that entail a chunk.

Now, the projecting function maps the first two distinct atomic sequential units separately as $X$ and $Y$, i.e. $f(A) \to X$, $f(B) \to Y$, $f(C) \to X$, $f(D) \to Y$, $f(E) \to X$, $f(F) \to Y$.

When this projection function is applied to every single chunk, then $f(ABBAAABBB)$, $f(CDDCCCDDD)$ and $f(EFFEEEFFF)$ all map to the same sequence $XYYXXXYYY$ in the projected space. The probability of observing such sequence in the projected space sums up the individual sequences in the observational space $P(XYYXXXYYY) = P(ABBAAABBB) + P(CDDCCCDDD) + P(EFFEEEFFF)$.

### K.2   Simulation Demonstration

We show a simulation to demonstrate that HCM learns motifs. In this experiment, HCM learns 40 trials of sequences. Each sequence is 12 units in length. There is an underlying structure in the motif space that generates sequences of every trial. In each trial, the sequence is made up of letters sampled from the set $\{R, B, G, T, P, Y\}$. Shown in Figure 11 as an example, sequences generated from a projected motif space can be BBBGBGBGGGGB as the first trial, and RRRYRYRYYYYR as the second trial, etc.

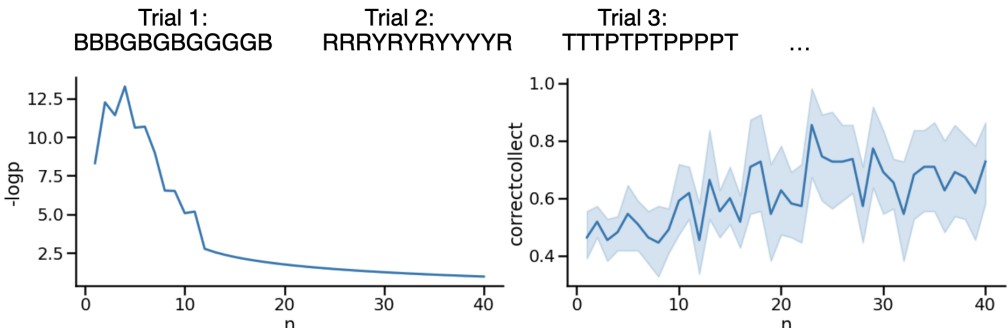

Figure 11: Learning projected motif chunks.

HCM learns from these sequences one after another. In each trial, the projecting function that maps the first two distinct elements to $X$ and $Y$ is applied to the sequence. HCM gradually learns and constructs motifs composed of $X$s and $Y$s in the projected space by constructing atomic units in the motif space and combining motifs together to bigger motifs. This way, HCM learns a belief set $\mathbb{B}$ of motifs. Each motif contains an estimated probability of occurrence, and thereby enables the evaluation of sequence information content $-\log p$ calculated by the probability of motifs that are used to parse a sequence, when the sequence is displayed to HCM. Smaller $-\log p$ implies less information content in the sequence.

The left plot of Figure 11 shows the information content contained in every sequential trial. The right plot of Figure 11 shows the imagination accuracy, which is the case that when HCM generates a sequence with the same length, the percentage of agreement with the sequence with such underlying motif. Both plots show convergence as the number of trials $n$ increases. Within 40 trials of sequences, HCM learns the underlying structure generating the sequence. Furthermore, given a novel sequence such as $TTTPTPTPPPPT$, the observation of $T$ and $P$ as distinct entities enable HCM to predict the entire sequence after observing the only the first 4 elements, even if this sequence has never appeared before.

Code and data for the experiments are available with this link: `https://github.com/swu32/HCM`