# OpenReview forum: "Learning Structure from the Ground up---Hierarchical Representation Learning by Chunking"
_NeurIPS.cc/2022/Conference — NeurIPS 2022 Accept_

### Official Review · Reviewer_2nnD · 2022-07-07

**Rating:** 5
**Confidence:** 4
**Soundness:** 3 good
**Presentation:** 2 fair
**Contribution:** 3 good

**Summary:**

The authors give a statistical algorithm for hierarchical chunking of data tokens.  They prove that when their algorithm's structure matches an underlying data-generating process, it can and will learn the correct structure.  They then demonstrate the interpretability of the learned chunkings, with application to cognitive science.  They perform simple experiments with visual, temporal, and visuotemporal chunking, and demonstrate that an application of their algorithm to fMRI data finds chunks corresponding to known areas of the brain.

**Questions:**

When the authors write about the Gestalt principle of grouping by proximity, is this meant to reflect proximity in topographic neural maps, egocentric space, or allocentric space?

What is the big difference between learning a semi-Markov HMM (Hidden Markov model) or hierarchical HMM with a likelihood exploiting the Size Principle (https://www.sciencedirect.com/science/article/abs/pii/S0001691809001553) and the actual HCM?

Do the independence tests used in the HCM have a relation to an optimization criterion, such as total correlation or similar?  Why use p<0.05, since we generally want stronger guarantees in an automated system?

What is the beta-test statistic mentioned on line 178-179?

What did the authors mean by an "overlapping representation" in terms of switching environments at line 187?  Representations are typically a part of some internal model, not a part of the environment itself.

**Limitations:**

The authors have mostly addressed my questions and concerns, but nonetheless, using a false-positive rate of 5% when you expect to form more than 20 chunks means you expect there to be at least one false-positive chunk.

**Strengths And Weaknesses:**

Strengths:
* Originality and clarity of the statistical justification for chunking
* Strong experiment comparing the HCM to an RNN for predictive convergence

Weakness:
* Incorrect use of p-values
* Too much of how the HCM actually works is in the appendix: pseudo-code, halting criteria, etc.

---

> ### Author Response · Authors · 2022-08-02
> **Author Response to Reviewer 2nnD (1/5)**
>
> Dear Reviewer 2nnD,
>
> Thank you very much for reviewing our paper. We are glad to hear that you found our work to be "original" and "clearly written" and that you think our experiments are strong. Please find a detailed response to your comments below.
>
> *The actual algorithm is entirely bottom-up and appears somewhat ad-hoc*
>
> Thank you for raising this point. Indeed, our algorithm is bottom-up: HCM learns rich representations from non-iid data by starting out only with atomic units. This makes it a one-size-fits-all method that does not demand any domain expertise to designate the structure of the model to fit the data. Rather, HCM discovers the structure within the data by itself. The learned representations are meaningful and interpretable. In future work, one could combine HCM with more top-down algorithms, for example, by pre-training on other sequences similar to what we have done in our experiments on transfer. We will also add the following paragraph on this possibility to our discussion:
>
> >HCM also opens up other application directions. One direction is integrating HCM with deep neural network approaches as an interface between human understanding and distributed computation. Learning hierarchies of coherent activations from intermediate hidden units has the potential to reveal neural networks' underlying computation structure. ***Furthermore, it is also possible to equip HCM with additional top-down encoded representations, for example, by pre-training on other sequences or by adjusting the chunks by hand before the training starts.***
>
> We are also open to further suggestions for inductive biases or other features of top-down algorithms that are expected to be useful across domains.  Please note that most popular sequence models are also largely bottom-up, relying on few structural constraints (for example, RNNs).
>
> Furthermore, we do not think that the algorithm is ad-hoc. Instead, it is based on principled assumptions taken from cognitive science that items that appear together frequently should form a unit, which is commonly called a chunk, which is a rational way of learning representations when the sequential data contain embedded hierarchies. We would be happy though to discuss further the justifications of our modeling decisions in which way our algorithm might be ad hoc, and add that discussion to the final paragraphs of our paper. For example, we are currently using the same algorithms for all of the presented problems and did not adapt the algorithm to any problem in particular, which is frequently the case for other algorithms. We also follow standard statistical conventions and set the value of the independence criterion to “p<0.05”. We will discuss this choice in more detail in our response to that particular point.
>
> *Too much of how the HCM actually works is in the appendix: pseudo-code, halting criteria, etc.*
>
> We thank the reviewer for pointing this out. Because the pseudo-code occupies a substantial amount of space, we are not able to move it to the main text without exceeding the current page limit. However, we now refer to the pseudo-code more directly in our updated manuscript. Moreover, we added the following paragraph describing the halting criterion to the main text:
>
> >From these dependent chunk pairs, the pair with the largest estimated joint probability is combined into a new chunk. The new chunk enlarges the belief set by one. The chunks in the new belief set are used to parse the sequence in the next iteration. This process repeats until all of the chunks in the belief set pass an independence halting criterion, which measures if all of the chunks in the belief set are currently independent, again assessed via a $\chi^2$ test (see SI).
>
> Please let us know if you think this is enough or if there is any other information that you would like us to add to the main text (while considering the page limit).

---

> > ### Author Response · Authors · 2022-08-02
> > **Author Response to Reviewer 2nnD (2/5)**
> >
> > Dear Reviewer 2nnD,
> >
> > Please find a continuation of our response to your questions and comments below:
> >
> > *Chunking low-dimensional GIFs doesn't seem very applicable to the real world.*
> >
> > The reason for running our model on GIF data is to demonstrate the possibility of learning representations of independent objects separated by the background by learning coherent chunks of observational units. In Figure 4, we show that HCM extracts the squid as an entity separated from the rising bubbles. This is relevant for inferring the presence of partially occluded objects.
> >
> > We agree that chunking a 25 x 25-pixel dimension GIF sequence is not directly applicable to some of the data in the real world, such as video data with a million pixels. However, there are cases where researchers are interested in returning representations for low-dimensional videos, for example, a specific range of ROI in the brain or tagged behavioral sequences in data. In the discussion, we point this out as a limitation and suggest methods that allow increasing the dimensionality of the input data:
> >
> > >Our work has its limitations. At the moment, HCM learns representations from semi-high dimensional sequential data (i.e., currently between 1 to 625 dimensions). We are actively looking into generalizing this algorithm to higher dimensional data domains by combining it with existing neural network approaches or computer vision algorithms such as coherent point drift (Myronenko and Song 2010) or normalized cuts (Shi and Malik 2000) to allow for the learning of ambiguous and high dimensional chunk exemplars. It is also possible to combine HCM with the compressed representation, such as the hidden activity of an auto-encoder to process and learn the structure from downstream representation.
> >
> > *When the authors write about the Gestalt principle of grouping by proximity, is this meant to reflect proximity in topographic neural maps, egocentric space, or allocentric space?*
> >
> > Thank you for asking this question. The Gestalt principle of grouping by proximity describes grouping behavior in egocentric space, i.e., the assumption that people tend to perceive things that appear close to each other together as a unit.
> >
> > The idea is that the Gestalt grouping principle exerted by human cognition is a rational way of learning from sequential data with an underlying hierarchical structure and that this way can be applied to several domains of processing and making sense of non-iid data.
> >
> > Therefore, the main message from the paper is that the gestalt principle of grouping is not only a principle in cognition but rather a principle to processing any type of such sequential data, regardless of observations in egocentric space, allocentric space (videos, movements of animals), or topographic neural maps.
> >
> > We are sorry that this message was not conveyed more clearly in our initial submission. Therefore, we decided to change some paragraphs of the introduction to address this question:
> >
> > >HCM starts out learning a minimal set of units sufficient to explain the sequence and gradually combines these units into increasingly larger and more complex chunks, constructing interpretable hierarchical structures. We derive learning guarantees on an idealized generative model and demonstrate convergence on sequential data coming from this generative model. **Thereby, Gestalt principles of grouping can be understood as a rational way of learning representations from sequences with an inherent hierarchical structure.** We then show that HCM resembles human learning in qualitative ways (compared to a recurrent neural network): it resembles human chunk learning and reaction time measures and flexibly transfers components learned from one task to another. We extend HCM to the visual-temporal domain capable of learning visual-temporal parts and wholes from higher dimensional sequential data. Taking it one step further, we deploy HCM to learn from fMRI data, which exerts a hierarchical structure. We demonstrate HCM's interpretable feature extraction ability to discover submodules of brain activations directly linkable to behavior confirmed by the neuroscientific literature and discoveries such as the correlation between brain chunk size with age and stimulus-specific activation chunks of brain areas.

---

> > > ### Author Response · Authors · 2022-08-02
> > > **Author Response to Reviewer 2nnD (3/5)**
> > >
> > > Dear Reviewer 2nnD,
> > >
> > > Hereby we continue to address your comments. Sorry about the continuation of postings. We want to address your questions and concerns systematically, but each official comment is limited to 5000 characters. Please find a continuation of our response below:
> > >
> > > *What is the big difference between learning a semi-Markov HMM (Hidden Markov model) or hierarchical HMM with a likelihood exploiting the Size Principle (https://www.sciencedirect.com/science/article/abs/pii/S0001691809001553) and the actual HCM?*
> > >
> > > HMMs are based on the assumption of a pre-defined structure in the data generation process and subsequently fit the model parameters to the data. Hierarchical HMMs extend HMMs to be able to capture multi-scale sequential structures. Semi-Markov HMMs assume that there is a hidden state, and the change of that hidden state depends on elapsed time. Although these models can capture hierarchical sequential structure, they lack HCM’s ability of adaptive recombination and reuse of pre-existing components.
> > >
> > > Our understanding of how to apply the size principle to HMMs, in this case, would be to inform when to create a new hidden state/chunk. How this would differ from our approach depends on how the size principle is applied here. Was there a specific approach the reviewer had in mind? Nonetheless, inference in HMMs can be challenging – particularly with non-parametric priors like the size principle. Our approach (by being entirely bottom-up) avoids some of these challenges.
> > >
> > > In short, HMMs and variants of HMMs need to specify a structure such as the number of states, hierarchy and waiting time (or nonparametric priors over these) before fitting these models to data, while no such assumptions are required need to be made for HCM. HCM discovers patterns and constructs structure from data. Nonetheless, we agree with the reviewer that it is important to distinguish HCM from HMMs and other related models. Hence, we will add the following paragraph to our related work section:
> > >
> > > > Another category of models to learn from sequential data includes Hidden Markov Models (HMMs) and their variants such as Hidden semi-Markov model (Yu 2015) and hierarchical HMMs (Fine et al. 1998), exploiting the size principle (Navarro and Perfors 2010) to capture multi-scale sequential structure. These models demand a pre-specification of structure before fitting their parameters to the data. They also lack HCM’s adaptive recombination and reuse ability.
> > >
> > > *Do the independence tests used in the HCM have a relation to an optimization criterion, such as total correlation or similar? Why use p<0.05, since we generally want stronger guarantees in an automated system?*
> > >
> > > We adhered to the standard convention in statistics of a level of significance set to 0.05. This parameter could be further adjusted to trade off deliberately between type 1 and type 2 errors when using HCM in different domains. For example, for medical data, you might want to have only a few chunks and should, therefore, set alpha to be conservatively low. However, when using chunks for predictions in downstream tasks, you might want to have more of them and should, therefore, set alpha to be liberally high.
> > >
> > > We will add the following paragraph to the supplementary section further describing the criterion:
> > >
> > > >We chose 0.05 as a decision criterion for rejecting the null hypothesis of two consecutively occurring chunks to be independent to be consistent with standard conventions in statistics. However, in applications, the strictness of parameters can be adapted to task domains.  For example, for medical data, you might want to have only a few chunks and should, therefore, set alpha to be conservatively low. However, when using chunks for predictions in downstream tasks, you might want to have more of them and should, therefore, set alpha to be liberally high.
> > >
> > > *What is the beta-test statistic mentioned on line 178-179?*
> > >
> > > Thank you very much for this question. Beta is a standardized regression coefficient and can therefore be interpreted as an effect size of the regression models relating the scaled log probability of HCM or RNN’s predictions to human RTs. We have further clarified this in the main text and also added a direct model comparison between these two regressions (HCM on RTs vs. RNN on RTs) using the Bayesian Information Criterion:
> > >
> > > > Yet, the relationship was substantially stronger between HCM ($\beta = 16.74$, $p \leq 0.001$, rank correlation $\tau=0.165$, $BIC=313586.5$) and human participants compared to that of the RNN ($\beta = 9.24$, $p \leq 0.001$,  $\tau=0.085$, $BIC=314236.4$). These results suggest that HCM resembles human chunk learning more strongly than RNNs and can therefore be seen as the cognitively more plausible approach to hierarchical representation learning.

---

> > > > ### Author Response · Authors · 2022-08-02
> > > > **Author Response to Reviewer 2nnD (4/5)**
> > > >
> > > > Dear Reviewer 2nnD,
> > > >
> > > > Hereby is a continuation of our response to your comments:
> > > >
> > > > *What did the authors mean by an "overlapping representation" in terms of switching environments at line 187? Representations are typically a part of some internal model, not a part of the environment itself.*
> > > >
> > > > We have now further clarified the distinction between a representation acquired by HCM and an overlapping structure in the generating model. We mean that the environment is a sequence coming from a generative model that contains overlapping structures, i.e., true underlying chunks, compared with the learned representation by HCM. An overlapping representation occurs when HCM is trained on an environment with an overlapping structure as compared to the test environment later on. We have adapted the caption of Figure 4 to mark this difference better:
> > > >
> > > > > a: Example of a representation learned by an HCM. b: A facilitative environment with a generative model overlapping in its structure with the test environment. Gray shadows mark the chunks that can be directly transferred. c: Average performance over the first 500 trials after the environment switches. d: Interfering environment. Gray shadows marks chunks that need to be acquired anew. e: Average performance over the first 500 trials after the environment switches.
> > > >
> > > > In the main text, we describe the representations only as something that has been learned by a model:
> > > >
> > > > >One characteristic of human learning is that previous learning experience facilitates and sometimes interferes with acquiring a new skill (Jarvis and Pavlenko 2007). Having an interpretable representation can inform us about positive or negative transfer a priori.
> > > >
> > > > >An HCM has learned a chunking graph in Figure 4a from an environment. When it switches to another environment with a generative model overlapping with its previously acquired representation, it can reuse the learned subgraphs of chunks marked in gray as in Figure 4b and learns faster than a naive HCM in Figure 4c. Vice versa, transfer can be detrimental when there is no or little overlap between the learned chunking graph and the generative model of the new environment. For the same HCM as in Figure 4 a, transferring to an environment with a chunking graph in Figure 4d implies learning the shaded chunks anew, in addition to running the risk of being misled by previously learned representations. As a result, the early performance of the pre-trained HCM suffers more from an interfering environment than a naive model in Figure 4e. Interpretability of HCM's representations enables the assessment of facilitation or interference when the environment changes.

---

> > > > > ### Author Response · Authors · 2022-08-02
> > > > > **Author Response to Reviewer 2nnD (5/5)**
> > > > >
> > > > > Dear Reviewer 2nnD,
> > > > > Here is our response to your last comment:
> > > > >
> > > > > *The authors' neuroscience experiment should be considered a limitation. The insula and cingulate are so connected that practically any measure of relation will detect their relationship, and so I'd prefer to see some less obvious findings.*
> > > > >
> > > > > We thank the reviewer for this comment. However, we do not think that the current results should be considered a limitation for the following five reasons:
> > > > > 1. The first thing we tried was to validate our model by looking into the literature on brain area connectivities. By definition, the relation between the brain areas has to be known for the chunk to be a good validation example. That insula and cingulate are connected is a good sanity check of our model, even if it is obvious.
> > > > > 2. However, this is not the only connection our algorithm found. For example, we also found the coactivation of right DLPFC, left DLPFC, right Frontal Pole, right Posterior Temporal Cortex, and left Parietal Cortex together as a chunk, which are sometimes seen to belong to the Visual Attention Network. Please also see the SI Section F.6.1 for more behaviorally tagged example chunks. Many of these examples are not obvious and may not correspond to known findings. We do acknowledge that the choice of a simple dataset and atlas is not perfect, but it suffices to show that HCM discovers interesting patterns in brain activation data.
> > > > > 3. Another intriguing feature of HCM is the nested chunk structure that it returns. We have put more examples in SI Section F.6.1. Essentially, these higher-order chunks activate together during the recording session, and so do the activations of chunks embedded in the hierarchy. This relates to the long-standing hypothesis on the nested network structure of the brain, with findings such as in [Boldassano 2017](https://www.sciencedirect.com/science/article/pii/S0896627317305937).
> > > > > 4. We also show the usefulness of our algorithm for further analysis. For example, we find that there are behavioral-specific chunks that are more likely to be activated when one particular type of scene is displayed in the movie.
> > > > > 5. Finally, we found a correlation between average chunk size and age. This is a novel finding in line with the reported findings in the original paper from which we adapted the data.
> > > > >
> > > > >
> > > > >
> > > > > In summary, we do not think that the fMRI results constitute a weakness of our paper but instead that they show one intriguing example of learned representation from rich, non-iid data. We hope that our method will help researchers in the neuroscience community to discover patterns inherent in neural activities.

---

### Official Review · Reviewer_SQe7 · 2022-07-11

**Rating:** 8
**Confidence:** 4
**Soundness:** 3 good
**Presentation:** 3 good
**Contribution:** 3 good

**Summary:**

The authors proposed and tested a Hierarchical Chunking Model. This model discovers a hidden hierarchical structure while having a stream of state transitions as inputs. By processing a linear stream of one-dimensional information as transitions between sets of different lengths of sequences, the model starts to build a hierarchical structure. Human participants also use this chucking strategy. The hierarchical structure inferred from changes in the reaction times is closely related to the model-generated structure. Last, the authors provide various examples in which sequential information processing can get benefit from this human-like chunking strategy.

**Questions:**

Is there any threshold determining which level of correlation is good enough to have a new chunk? And how the memory decay parameter and the deletion threshold are determined? I may miss how they were determined. Are they predetermined as fixed values? Or, these can also be determined by another hierarchical variable which tracks to what extent the environment is volatile. In this way, the model may be able to reduce the learning time.




**Limitations:**

The current model may not be able to generalize the structure learned from a sequence if the stimuli changes while keeping the same structure. (e.g., 12221212 -> 34443434).

**Strengths And Weaknesses:**

Strengths. By constructing a hierarchical structure, the model can efficiently learn a novel sequence if it was generated from the same structure and afford generalization. This hierarchical chunking model doesn't require a prior model structure but can be generated by the data itself. Interestingly, without knowing the outer stimuli causing these activations, this model could capture the meaningful neural networks associated with different cognitive states.

Potential weaknesses. The model only learns the sequence determined by the deterministic transition. This may capture better some types of sequences (where a temporal order matters). However, this can be more flexible if the model can learn the hidden structure generating a chunk of sequence. For example, if a sequence is generated by a random walk, the current model would increase the number of independent chunks, while it could be better to be captured by a certain network structure describing the states of each of the communities and how they are connected.

---

> ### Author Response · Authors · 2022-08-02
> **Author Response to Reviewer SQe7 (1/2)**
>
> Dear Reviewer SQe7,
>
> Thank you for your feedback and suggestions. We are glad that you found the model "interesting", "efficient" and appreciated the data-driven prospect of constructing representations. We have addressed your comments and questions below and are looking forward to engaging with you further.
>
> *Potential weaknesses. The model only learns the sequence determined by the deterministic transition. This may capture better some types of sequences (where a temporal order matters).*
>
> The model does not learn by deterministic transitions but is probabilistic and can learn chunks with non-deterministic transitions. And a sensitivity analysis shows that a temporal order or phase shift, or noise does not systematically affect learning convergence, see SI Section I on Sensitivity Analysis.
>
> *However, this can be more flexible if the model can learn the hidden structure generating a chunk of sequence. For example, if a sequence is generated by a random walk, the current model would increase the number of independent chunks, while it could be better to be captured by a certain network structure describing the states of each of the communities and how they are connected.*
>
> This is what our model is doing, i.e., it is learning the hidden structure and generating a sequence of chunks. One can show that a random walk on a network can be re-expressed as a transition matrix determining the probabilities of movements between the network nodes. Learning this structure is de facto what we set the model up to do. Please let us know if you had something in particular in mind when mentioning network structure. Do you mean that HCM currently cannot abstract away from transition probabilities?
>
> *Is there any threshold determining which level of correlation is good enough to have a new chunk?*
>
> Chunks are determined by using an independence testing criterion. We set the probability of this test as when a transition is strong enough to be chunked together to 0.05 or lower and prefer to create new chunks with higher evidence of non-independence. However, this parameter can be tampered with. See the discussion about type 1 and type 2 errors in response to the other reviewers.
>
> *And how the memory decay parameter and the deletion threshold are determined? I may miss how they were determined. Are they predetermined as fixed values? Or, these can also be determined by another hierarchical variable which tracks to what extent the environment is volatile. In this way, the model may be able to reduce the learning time.*
>
> We thank the reviewer for this comment. Currently, the memory decay parameter and the threshold were set between 0.9 and 1 for memory decay and 0.01 for the deletion threshold. We did not tune these parameters to the tasks at hand but rather set the to a priori plausible values. The reviewer is correct in pointing out that memory decay changes the adaptability of the environmental dynamics of the sequence, and that determining them adaptively could further improve HCM, especially in volatile environments. We have added the following paragraph to the discussion acknowledging this possibility:
>
> > Currently, we fix the memory decay and the deletion threshold parameters to a priori plausible values. In future work, these parameters could be adapted online, further improving HCM’s performance, especially in volatile environments.

---

> > ### Author Response · Authors · 2022-08-02
> > **Author Response to Reviewer SQe7 (2/2)**
> >
> > Dear Reviewer SQe7,
> >
> > Please find the response to the rest of your comments below:
> >
> > *Limitations: The current model may not be able to generalize the structure learned from a sequence if the stimuli changes while keeping the same structure. (e.g., 12221212 -> 34443434).*
> >
> > Thank you for pointing out this possibility. This comment has prompted us to consider how HCM could learn about chunks if the structure changes as described by the reviewer. One way to do this is to learn chunks in a projective space, akin to learning motifs in sequential data. We have sketched out a detailed approach to how this could be done in a section on “Learning Motifs” in SI Section K. Briefly, one needs a projecting function that maps chunks into a projective space and then runs HCM over this projective space. In this way, the sequences 12221212  and 34443434 will both be parsed as XYYYXYXY in the projected space. HCM learns chunks in this example by constructing XY first as a chunk, followed by learning the XYXY, then YY …etc. As the number of observations increases for each sequence, HCM gradually learns the chunk XYYYXYXY. Furthermore, given a novel sequence such as 56665656, HCM can readily generalize from the first two sequential elements to predict the rest of the sequence, even when this sequence never appears in the training, without loss of accuracy.
> >
> > We also mention this possibility in the Discussion section of our main paper:
> >
> >
> > > We show that HCM can be generalized to not only learn simple chunks but also chunks in projected spaces and thereby generalize between two chunks that contain the same motif (for example, 12221212  and 34443434; see SI for detailed results). In the future, it might be worthwhile to further combine our approach with others amongst a taxonomy of representational hierarchies.

---

> ### Comment · Reviewer_SQe7 · 2022-08-04
> **Changed my ratings after having authors' responses**
>
> I appreciated that the authors provided clarifications to my questions (while including these additional explanations and rationales in the supplementary information) and additional substantive simulations (especially sections I and K) which helped a lot to correct my initial misconception.
>
> I changed my rating in Presentation to 3, the overall Rating to 8, and my Confidence level to 4.

---

> > ### Author Response · Authors · 2022-08-08
> > **Thank you**
> >
> > Dear Reviewer SQe7,
> >
> > Thank you for taking the time to read the updated manuscript. We are glad that the additional simulations helped with the clarification. We greatly appreciate your questions and constructive suggestions to make this work better.

---

### Official Review · Reviewer_ybpV · 2022-07-11

**Rating:** 6
**Confidence:** 4
**Soundness:** 3 good
**Presentation:** 4 excellent
**Contribution:** 2 fair

**Summary:**

This work proposes a model that learns hierarchical chunks from non-i.i.d data in an unsupervised fashion. The central part of the model uses a statistical test to assess the dependence between consecutive items in a sequence. The authors use a sequence learning task to show that the model learns more similarly to humans than RNNs, and predicts human reaction times more strongly. This leads them to conclude that the HCM is more cognitively plausible than an RNN. They also show the performance of the model on visual sequences and resting state fMRI data, and illustrate the ability of the model to do online learning.

**Questions:**

* What are the memory demands of the model? A formal analysis would be useful, given the vague claim that the HCM can handle "semi-high dimensional" data. This is also of interest because one of the uses for chunking may be in memory compression.
* How did you choose the size and architecture of the RNN for comparison with the HCM?
* The text suggests that the online HCM can be subject to memory decay to adapt to changing statistics. Is a significant decay / deletion threshold actually required for any of the present experiments?
* Supplemental Section G gives a hyperparameter for the HCM reaction time analysis (the probability of within chunk RT is 1 - 4/eps). Why was this value chosen? What happens when other values are chosen?
* What are the implications of actual value of the statistical threshold? Is it possible that different contexts may lead learners to have more strict or relaxed thresholds?

Suggestions (from major to minor)
* Dedicate a separate section to Related Work.
* Figure 1 does not define the variable d as depth.
* In the Supplemental Material E.1.3, Equations 23 & 24 appear to be identical.

**Limitations:**

Edit: the authors have addressed these items on the checklist.

The authors did not complete items 5a-5c on the checklist. I will rate this paper a reject as the ethical concerns are not completely addressed.

**Strengths And Weaknesses:**

Edit: I thank the authors for their extensive replies to the reviews. They have addressed several concerns, particularly with their proposed  changes to summarize related work in reply to both me and reviewer TVcV. For now I have increased my score to a Weak Accept (6). This may change with any continued discussion.

The experiments are varied and illustrate different possible uses of the model. In particular, the extension of the HCM to spatiotemporal data and to online learning are both intriguing areas of future research. I also appreciate the pursuit of cognitively plausible models, the proof of recoverability, and the discussion of the importance of hierarchical representation learning in human intelligence.  The paper and method are also explained clearly, though some details about experimental choices are missing (even when referring to the supplement).

I do find the discussion of how this model relates to other previous approaches quite lacking. This makes the originality and significance of the proposal less clear, as some related work is missing. They write that "other types of hierarchical representations such as clustering or relational hierarchies are out of the scope for this work." Some clustering algorithms are certainly related and within scope. In NLP models, tokenizer methods such as Byte-Pair Encoding or WordPiece can be thought of as clustering approaches that, like chunking, facilitate compression of sequential data. Furthermore, it would be useful for the authors to discuss the relationship between their definition of a hierarchical structured sequence and a probabilistic context free grammar. They look quite similar to me, and much theoretical work has been done to study the ability of various models to capture different types of artificial languages (e.g. Hewitt et al. 2020, RNNs can generate bounded hierarchical languages with optimal memory). This discussion may even facilitate the authors' comparison of the HCM to other models.

Given this and some of the lacking details, I am rating the contribution of this paper as fair. If the ethical concerns are addressed, I would rate this as a 5 (borderline accept). I do think the framing of this work to connect it to the cognitive science literature on chunking is useful, and I am willing to increase this score if my concerns and questions are addressed.

---

> ### Author Response · Authors · 2022-08-02
> **Author Response to Reviewer ybpV (1/3)**
>
>
> Dear Reviewer ybpV,
>
> Thank you very much for your helpful feedback on our paper. We are glad to hear that you liked our model, in particular, the spatial-temporal components and the connection to cognitive science, as well as our proof of recoverability and experiments. We have addressed your concerns below and modified the manuscript according to your suggestions.
>
> *The authors did not complete items 5a-5c on the checklist. I will rate this paper a reject as the ethical concerns are not completely addressed. …*
>
> We understand that the reviewer rated our paper as “reject” because we did not complete items 5a-5c on the checklist and that the reviewer would have otherwise rated our paper as a 5 or above.
>
> We thank the reviewer for pointing this out. We appreciate your concerns about the ethical treatment of participants. Because there are no human experiments conducted in correspondence to this paper (we reanalyzed publicly available data), we left the question boxes blank and referred to the corresponding paper in the main text, for which we apologize. We have modified the checklist to the following:
>
> >5. If you used crowdsourcing or conducted research with human participants…
>
> >(a) Did you include the full text of instructions given to participants and screenshots, if
> applicable? [N/A] We used existing data from experiments conducted in previous
> work. This information can be found in the original articles.
>
> >(b) Did you describe any potential participant risks, with links to Institutional Review
> Board (IRB) approvals, if applicable? [N/A] This question is not applicable to this
> project.
>
> >(c) Did you include the estimated hourly wage paid to participants and the total amount
> spent on participant compensation? [N/A] Since no human experiment was conducted for
> this project and we used experimental data from other published and publicly available
> work, this question does not apply to us here and can be found in the original articles.
>
> *Related work section*
>
> We will add a related work section to the camera-ready manuscript to discuss how our model relates to other accounts, including the mentioned clustering models, HMMs, and context-free grammar. We particularly mention Byte-Pair-Encoding, WordPiece, as well as the study by Hewitt comparing RNNs and context-free grammar. Please note that we will only be able to add this to the camera-ready version because of the current page limitations. We have copied the proposed new “Related Work” section below:
>
> > Markov models, hidden Markov models, as well as n-gram models are traditional sequence models that are notoriously memory inefficient when it comes to processing sequences with high-order dependencies and long chunks. The parameters of these models proliferate exponentially as a function of chunk length because they do not apply a merging principle akin to that of HCM (see SI section G).
>
> > The principle of iterative merging of chunks has been used in compression algorithms, such as tokenizing methods in NLP, which were developed to optimize sequential data compression. Tokenizing methods such as Byte-Pair Encoding iteratively merges the most frequent pairs of chunks to build a vocabulary of a text corpus. This objective is easy to compute but gives rise to ambiguous parses of the text (e.g. [AB, C] / [A, BC]). To minimize parse ambiguity, WordPiece (Wu et. al. 2016) merges chunks that increase the likelihood of the corpus the most. However, the objective of WordPiece is extremely expensive to compute. HCM circumvents the problem of computing the global sequence likelihood by instead maximizing the local chunk continuation likelihood. The computational efficiency of HCM makes it a plausible cognitive model of chunking as well as a promising method for NLP.
>
> > Probabilistic context-free grammars (PCFGs) are related to HCM in that they use trees as a representational form of sequences. However, the parse trees of PCFGs denote production rules, such as S → NP + VP. These production rules define how abstract syntactic units (non-terminals), such as a noun phrase and a verb phrase, are instantiated into a concrete string of words (terminals) to compose a sentence, such as ‘ we wrote the paper’. In comparison, the generative tree of HCM denotes statistical relationships among concrete chunks, such as ‘we’-‘wrote’-’the paper’. Extending HCM to represent abstraction is an exciting future direction, on which avenue the comparison to PCFGs will be instructive.

---

> > ### Author Response · Authors · 2022-08-02
> > **Author Response to Reviewer ybpV (2/3)**
> >
> > Dear Reviewer ybpV,
> >
> > Please find a continuation of the response to your comments and questions below:
> >
> > *What are the memory demands of the model? A formal analysis would be useful, given the vague claim that the HCM can handle "semi-high dimensional" data. This is also of interest because one of the uses for chunking may be in memory compression.*
> >
> > We have added two additional sections about HCM’s memory demands to the SI sections G and H. SI section G translates the hierarchical graph learned by HCM into an nth order Markov Model and evaluates the upper bound on the order of the Markov Chain to capture an equivalent hierarchical chunking graph learned by HCM. We find that the number of states for an equivalent Markov chain for the representation learned by HCM is bounded by $|\mathbb{B}|\times 2^d$, where $d$ is the depth of the hierarchical graph and $|\mathbb{B}|$ is the size of the belief set. This upper bound grows exponentially with depth.
> >
> > SI Section H on memory analysis formulates a measure of Expected Unit Explanatory Power, a measurement of the average lengths of the sequence spent per bit of information to store chunks in the belief set, and walks the reader through an example showing how chunks are efficient for encoding sequences.
> >
> > *How did you choose the size and architecture of the RNN for comparison with the HCM?*
> >
> > This experiment was intended as a qualitative comparison of the chunk learning rate between humans and neural networks. Thus, we chose a generic RNN architecture with 40 embedding dimensions and three layers, each with 40 LSTM hidden units and a dropout rate of 0.2, followed by a fully connected feedforward layer.
> >
> > We compared the chunk size increase rate across three RNN models varying in sizes and added it as a separate section to the SI section F.5.1. The comparison with RNNs with half of the size or double of the size showed qualitatively similar learning trends. Additionally, we have included Figure 4 in SI showing the chunk size increase across RNNs with different architectures compared to humans. None of these RNNs led to human-like learning rates.
> >
> > *The text suggests that the online HCM can be subject to memory decay to adapt to changing statistics. Is a significant decay / deletion threshold actually required for any of the present experiments?*
> >
> > Memory decay ensures that HCM does not adhere to too many chunks or obsolete chunks and thereby enforces sparsity in its representations. In this way, memory decay benefits learning a sparse set of meaningful chunks that can be interpreted. Without the decay, the belief set would become increasingly larger and contain several chunks that rarely occur. Moreover, the memory decay parameter is helpful for scenarios when the generative model changes, as we have investigated in the transfer learning section. Learning about a new generative model and adapting the underlying representations efficiently would not be possible without memory decay.
> >
> > *Supplemental Section G gives a hyperparameter for the HCM reaction time analysis (the probability of within chunk RT is 1 - 4/eps). Why was this value chosen? What happens when other values are chosen?*
> >
> > We choose a small epsilon because the reaction time of within-chunk items is typically faster than between-chunk items (See [Koch and Hoffmann 2000](https://pubmed.ncbi.nlm.nih.gov/10743384/), [Gobet et. al. 2001](https://pubmed.ncbi.nlm.nih.gov/11390294/), [Vewey 1996](https://citeseerx.ist.psu.edu/viewdoc/download?doi=10.1.1.554.6990&rep=rep1&type=pdf)). Following the reviewer’s suggestion, we have also tried to set epsilon to other meaningful values: 0.01, 0.05, and 0.10. The resulting regression coefficients remained largely unchanged:
> >
> >
> > Estimate:
> >
> > $\beta$ (eps = 0.01) = 16.82
> >
> > $\beta$ (eps = 0.05)= 16.74 Original
> >
> > $\beta$ (eps = 0.10)= 16.60
> >
> >
> > Standard Error:
> >
> > SE (eps = 0.01) = 2.557
> >
> > SE (eps = 0.05) = 2.552 Original
> >
> > SE (eps = 0.10) = 2.543

---

> > > ### Author Response · Authors · 2022-08-02
> > > **Author Response to Reviewer ybpV (3/3)**
> > >
> > > Dear Reviewer ybpV,
> > >
> > > Please find the response to your last question below:
> > >
> > > *What are the implications of the actual value of the statistical threshold? Is it possible that different contexts may lead learners to have more strict or relaxed thresholds?*
> > >
> > > For p < 0.05, we used the standard convention in statistics to test the level of significance. This choice means that the probability of committing type I error is bounded by 0.05, which is the probability of chunking elements together when there are no correlations amongst the elements. Decreasing alpha will lead to more parsimonious chunking. The strictness of parameters should be adjusted according to application domains.  For example, for medical data, you might want to have only a few chunks and should, therefore, set alpha to be conservatively low. When using chunks for predictions in downstream tasks, you might want to have more of them, and should, therefore, set alpha to be liberally high.
> > >
> > > We have added the following paragraph to the SI further explaining how the threshold can be adapted:
> > >
> > > > We chose 0.05 as a decision criterion for rejecting the null hypothesis of two consecutively occurring chunks to be independent to be consistent with standard conventions in statistics. However, in applications, the strictness of parameters can be adapted to task domains.  For example, for medical data, you might want to have only a few chunks and should, therefore, set alpha to be conservatively low. However, when using chunks for predictions in downstream tasks, you might want to have more of them, and should, therefore, set alpha to be liberally high.

---

> > > ### Comment · Reviewer_ybpV · 2022-08-08
> > > **Memory decay parameter**
> > >
> > > > The text suggests that the online HCM can be subject to memory decay to adapt to changing statistics. Is a significant decay / deletion threshold actually required for any of the present experiments?
> > >
> > > >Memory decay ensures that HCM does not adhere to too many chunks or obsolete chunks and thereby enforces sparsity in its representations. In this way, memory decay benefits learning a sparse set of meaningful chunks that can be interpreted. Without the decay, the belief set would become increasingly larger and contain several chunks that rarely occur. Moreover, the memory decay parameter is helpful for scenarios when the generative model changes, as we have investigated in the transfer learning section. Learning about a new generative model and adapting the underlying representations efficiently would not be possible without memory decay.
> > >
> > > I was actually asking to know what value the decay parameter / deletion threshold was set to for various experiments. On a second read, I found this detail for the experiment transferring between environments in the supplement section F.2. Are these parameters set for the other experiments, e.g. the fMRI experiment or the comparison to human data / RNNs? I don't see them reported. The main text of the paper is vague on which experiments use this by simply saying "use the online model to learn representations in more realistic and complex set-ups."

---

> > > > ### Author Response · Authors · 2022-08-09
> > > > **Clarification on memory decay parameter**
> > > >
> > > > Thank you for the question. Indeed, these parameters are set for all of our experiments. We performed a parameter search for the transfer experiment and then just used the same sets of parameters (forgetting rate = 0.996, deletion threshold = 0.01) for the visual chunk learning, and the visual-temporal chunk learning experiments. For the human chunk learning experiment, we lowered the forgetting rate to 0.90 and increased the deletion threshold to 0.1 to be more in line with human forgetting (they are normally more forgetful than optimal agents). Finally, for the fmri experiment, we removed the forgetting process altogether (parameter of 1) because we wanted to have many chunks that could be used for the downstream tasks. We believe that changing the forgetting parameter does not affect our results much as long as the parameter is reasonably high (i.e. higher than 0.9).
> > > >
> > > > We have added the reported parameter values to the corresponding sections of the SI.

---

### Official Review · Reviewer_TVcV · 2022-07-11

**Rating:** 7
**Confidence:** 3
**Soundness:** 3 good
**Presentation:** 4 excellent
**Contribution:** 3 good

**Summary:**

This paper presents a hierarchical chunking model (HCM), which learns chunks from non-i.i.d. sequential data. The full learning process elicits a trace of building hierarchy from the ground up, similar to humans' perception and mind process, and hence human-interpretable. The proposed model is effective, adaptive, and successful in considered applications spanning from symbolic sequences to visual temporal data, from artificial settings to real-world problems like learning hierarchies of brain activation from resting-state fMRI data.

**Questions:**

1. Is it possible to have ambiguity in the process of parsing larger and larger chunks, e.g., what if there are multiple largest probabilities in the transition matrix (not necessarily an exact tie, but close)? Would one choice of breaking the (approximate) tie greatly change the later process? The whole procedure seems conceptually similar to hierarchical clustering, which merges small parts in a greedy fashion. Then, is the final result going to be sensitive to some "noisy" decisions made earlier? Here, a "noisy" decision may be a choice of breaking the tie as mentioned above. But, in addition to that, can the authors also explain if the final result tends to be sensitive to a small phase shift (e.g., start reading from the second item instead of the very beginning of a sequence)? Is it sensitive to noisy observations?

2. In Figure 2a, it seems 0 is treated differently from other symbols, why so and is this kind of treatment always needed for empty observation?

3. This paper presents several cases where the model is capable of discovering known patterns. While these are necessary to show the model's efficacy when tested against ground truth, has the model also exhibited its ability to discovery new patterns, e.g., in fMRI data?

**Limitations:**

Limitations and their connections to related and future work were adequately addressed.

**Strengths And Weaknesses:**

This paper is well written, whose ideas are easy to follow. The goal, motivation, and scope is clear; theory and algorithm parts are solid; experiments are well designed and conducted, spanning a diverse range of topics; experimental results are significant, which are further interpreted and discussed thoroughly. There are two things I would suggest the authors considering improving.

How the work presented in this paper is positioned among other related work in the literature is not clear to me until the final discussion section. However, comparisons were only discussed in general terms, largely focusing on the methodological perspective. If any of the techniques introduced in this paper are derived, influenced, or inspired by prior work in the similar field, could the authors include comparisons regarding performances on the considered experiments in the paper? The only experimental comparison in this paper seems to be with RNN (and human performance), which is methodologically quite different from the proposed HCM. In fact, I think comparing to RNN and showing HCM's superiority (e.g., as in Figures 2 and 3) is relatively weak, as HCM is especially built for the purpose of studying those internal sequential structures. For instance, the similar way how HCM and generative model in Sec 3 are built, should naturally make HCM work well for that type of generative model. So, to me, experimentally (rather than algorithmically) comparing HCM to more related other algorithms (like the ones mentioned in the discussion section) will be more convincing.

Figures are generally well designed, however, their layouts and arrangements are not that clean to me. For example, in Figure 1, it took me a while to figure out where the different subfigures are, and it seems to me there are overlaps between subfigures.

---

> ### Author Response · Authors · 2022-08-02
> **Author Response to Reviewer TVcV (1/3)**
>
> Dear Reviewer TVcV,
>
> Thank you very much for the helpful feedback and suggestions on our paper. We are glad that you found this paper well-written with solid theory parts supported by well-designed experiments with significant results addressing a range of topics. We have attempted to address your concerns below and modified the manuscript according to your suggestions.
>
> *How the work presented in this paper is positioned among other related work in the literature is not clear to me until the final discussion section.*
>
> We will move and add some comparisons with pre-existing literature from the discussion section to a new related work section (for the camera-ready version, where we can add an additional page), and incorporate models proposed by several of the reviewers so that the reader can have a clear understanding of how this paper is positioned among other related work:
>
> >HCM extends upon decades of previous cognitive science and psychology research on chunking. In cognitive science, process models such as PARSER and competitive chunking were demonstrated to generate qualitatively similar chunks as in human sequence learning (Perruchet and Vinter 1998, Servan-Schreiber and Anderson 1990). HCM is a rational algorithm that learns the underlying chunks when the sequence is generated from a hierarchical chunking graph. Therefore, the chunking criterion is no longer a heuristic but a rational learning strategy that enables hierarchical structural discovery. On top of inheriting the merits of its predecessors, HCM generalizes the chunk learning principle to higher dimensional sequential domains such as visual-temporal sequential data.
>
> >Another category of models to learn from sequential data includes Hidden Markov Models (HMM) and their variants such as Hidden semi-Markov model (Yu 2010) and hierarchical HMM (Fine et. al. 1998) to capture multi-scale sequential structure. These models demand a structure specification before fitting parameters to the data. They also lack the adaptive recombination and reuse of pre-existing components. The same issue is with neural network approaches to extract chunks from sequences (Si et. al. 2020, Ortmann 2021). Apart from lacking in interpretability, these models do not leverage the concatenation process observed in humans or reusing previously learned representations to construct new representations.
>
> *If any of the techniques introduced in this paper are derived, influenced, or inspired by prior work in the similar field, could the authors include comparisons regarding performances on the considered experiments in the paper? So, to me, experimentally (rather than algorithmically) comparing HCM to more related other algorithms (like the ones mentioned in the discussion section) will be more convincing.*
>
> Thank you for your suggestion. Following this suggestion, we compared our model (HCM) on a one-dimensional sequence with PARSER, which is the most related cognitive chunking algorithm, and added this comparison to SI Section J (Model Comparison with PARSER). In particular, we were interested in which model better describes human chunking.
>
> The chunk size increase between HCM, RNN, PARSER, and human behavior shows that HCM and PARSER continue to learn and build up longer chunks as they go through the sequence. Evaluating the average rate of chunk growth also showed that both PARSER and HCM are more similar to participants than the RNNs. The negative log-probabilities of sequence elements generated by the HCM, RNN, and PARSER were both significantly related to human reaction times. Yet the relationship was substantially stronger between HCM ($\beta = 16.74$, $p \leq 0.001$) and human participants than PARSER  ($\beta = 11.070$, $p \leq 0.001$) compared to that of the RNN ($\beta = 9.24$, $p \leq 0.001$).  In short, PARSER indeed matched human data better than RNNs, but not nearly as well as HCM.

---

> > ### Author Response · Authors · 2022-08-02
> > **Author Response to Reviewer TVcV (2/3)**
> >
> > Dear Reviewer TVcV,
> >
> > Please find a continuation of your comments and suggestions below.
> >
> > *Figures are generally well designed; however, their layouts and arrangements are not that clear to me. For example, in Figure 1, it took me a while to figure out where the different subfigures are, and it seems to me there are overlaps between subfigures.*
> >
> > Thank you for pointing this out. We have changed the layout and design of Figure 1 and other figures to make them more readable.
> >
> > *Is it possible to have ambiguity in the process of parsing larger and larger chunks, e.g., what if there are multiple largest probabilities in the transition matrix (not necessarily an exact tie, but close)? Would one choice of breaking the (approximate) tie greatly change the later process?*
> >
> > Theoretically, if the generative model contains chunks with tied probabilities, these chunks will be discovered in either one pair first or the other pair first by the rational HCM, in either of the tied ways, the order of chunking learned by the rational HCM shall be consistent with the order of chunks in the generative model, with the tied joint probabilities.
> >
> > Moreover, the probability of having exact ties in the transition frequencies rarely happens when the sequence gets long, and in the ideal case scenario with sequences of an infinite length, forming a tie is nearly impossible.
> >
> > *The whole procedure seems conceptually similar to hierarchical clustering, which merges small parts in a greedy fashion. Then, is the final result going to be sensitive to some "noisy" decisions made earlier?*
> >
> > If the model learns some chunks that are incorrect at the beginning, the chunks are not going to be used to parse the sequence and therefore will have smaller frequencies compared to some other chunks and be subject to memory decay. Therefore, starting with an incorrect representation does not affect the model to converge. One example of this can be seen in our transfer experiments (see SI Figure 2, Section F.2). Although at the beginning of the sequence, the model’s performance is not as good as a naive model’s performance, as the sequence length increases, the model still convergences.
> >
> > *But, in addition to that, can the authors also explain if the final result tends to be sensitive to a small phase shift (e.g., start reading from the second item instead of the very beginning of a sequence)?*
> >
> > Theoretically, HCM should not be sensitive to phase shifts as the phase does not affect how sequences are parsed. Nonetheless, we have run an additional simulation of learning representation from a hierarchical structure sequence with an increasing size of phase shift (starting from the nth item instead of the 1st) and added the results to SI Section I.2 (Phase Shift Analysis). These simulations confirmed that HCM is not sensitive to phase shifts.

---

> > > ### Author Response · Authors · 2022-08-02
> > > **Author Response to Reviewer TVcV (3/3)**
> > >
> > > Dear Reviewer TVcV,
> > >
> > > Here are our responses to the rest of your questions. Thank you for raising these questions.
> > >
> > > *Is it sensitive to noisy observations?*
> > >
> > > Noisy observation may affect the joint probability estimation, but not the order of it. Thus, the model’s learning ability should be robust when iid noise is added. To assess this intuition, we have run an additional simulation where we added noise to each element of a sequence. To simulate noisy observations, we added an $\epsilon$ probability of switching to an alternative atomic unit for each unit in the sequence. That is, for each sequential element, there was a $(1 - \epsilon)$ probability that this element stayed the same, and an $\epsilon$ probability of flipping to any other atomic unit with equal probability. We took an exponentially increasing level of $\epsilon$ ranging from $0$ to $0.1$, and trained HCM on these noise perturbed sequences. The results of these simulations showed that HCM is robust to noise and still converged in learning.
> > >
> > > *In Figure 2a, it seems 0 is treated differently from other symbols, why so and is this kind of treatment always needed for empty observation?*
> > >
> > > We introduced 0 to denote an empty observation in the generative model, which has the biggest probability mass to emulate the sparse observational signals of non-zero elements. However, for the subsequent datasets, 0 was no longer encoded as a symbol and HCM learned about the atomic units from scratch.
> > >
> > > *This paper presents several cases where the model is capable of discovering known patterns. While these are necessary to show the model's efficacy when tested against ground truth, has the model also exhibited its ability to discover new patterns, e.g., in fMRI data?*
> > >
> > > Thank you for asking this question. Indeed, many of the chunks discovered in the fMRI data are non-obvious and may not correspond to known findings. We have added more examples of scene-tagged brain activation chunks to SI Section F.6.1.
> > >
> > >
> > > Another intriguing feature of HCM is the nested chunk structure that it returned. We have put a couple more examples into the SI section F.6.1. Essentially, in these higher-order activations, bigger chunks activate during the recording session, and so do the chunks embedded in the hierarchy. This relates to the long-standing hypothesis about the nested network structure of the brain, with findings such as (e.g. https://www.sciencedirect.com/science/article/pii/S0896627317305937)
> > >
> > >
> > > Additionally, we find that there are behavioral-specific chunks that are more likely to activate when one particular type of scene was displayed in the movie. Finally, we also found a correlation between average chunk size and age, which has not previously been demonstrated. In summary, HCM was able to discover meaningful new patterns in the fMRI data.

---

### Review · Ethics_Reviewer_XuKD · 2022-08-05

**Recommendation:** N/A

**Ethics Review:**

The authors clarified that "no human experiment was conducted for this project."

---

### Review · Ethics_Reviewer_MEdY · 2022-08-22

**Recommendation:**

I think addressing the concerns won't be very hard for the authors, as the changes would be relatively small.

**Ethics Review:**

Apologies for the delay in the ethics review, this paper has been difficult to think through in terms of ethical concerns. My overall argument is that there are no ethical concerns but there are framing concerns.
This paper presents a method for learning longer chunks than what can be learned with an RNN. The authors build on this claim to argue that HCM resembles human chunking, however this framing I think does more harm than benefit. In particular, it reduces human pattern recognition to a matter of chunk learning, which I am sure that the authors do not believe (or at least their manuscript leads me to think that they don't believe this to be true).

So mostly, what I would propose is a small rewrite away from saying the HCM resembles human chunking but rather put the emphasis on that HCM learns longer chunks than RNNs, and in human experiments there is less of a gulf between the chunking HCM learns and the chunking patterns reported by humans.

The second concern that I have is the fMRI experiments, it's unclear whether the authors obtained IRB approval for this work. Moreover, the framing here seems very strong and I think the authors would benefit from softening the language a bit.

I would also suggest strengthening the limitations section, while this work does improve on chunking longer lengths, but there's little conversation around the likening to human comprehension, which ought to be addressed.

---

### Author Response · Authors · 2022-08-09
**Author Summary of Rebuttal Discussion**

First, we thank all four reviewers for their time and valuable feedback!

We appreciate the reviewers’ assessment of our work to have “originality and clarity” and “strong experiments” (2nnD); to be “well-written”, with a “clear goal, motivation and scope”, and thinking that our “theory and algorithm parts are solid”, “spanning a diverse range of topics” (TVcV); to involve “intriguing areas of future research” and be a “clearly explained paper with clear method” (ybpV); and with a response to the reviewers’ suggestions leading to “additional substantive simulations” (SQe7).

We received numerous helpful and constructive suggestions. This is a summary of the main concerns and how we addressed them:

- In response to reviewer TVcV, we ran a model comparison with PARSER as the most related model to HCM on human chunk learning data. HCM outperformed PARSER as a model of human chunk learning. Additionally, we conducted several sensitivity analyses showing HCM’s robustness to observational noise and phase shift.
- In response to reviewer TVcV and reviewer 2nnD, we added additional examples of fMRI chunks and chunk hierarchies that are more structured and do not correspond to known findings (added to the SI).
- In response to reviewer 2nnD, reviewer ybpV, and reviewer TVcV, we added a related work section to position our work better in relation to the literature, including a comparison with several variations of HMMs. We will add this section to the camera-ready version (where we will have one additional page available).
- In response to reviewer ybpV, we added two analyses on memory demand to the SI. The first relates HCM with Markov chains and the second explains the relation between chunking and memory compression. We have added a paragraph explaining the rationale for using a memory decay parameter and more details about parameter settings to the SI.
- In response to reviewer SQe7, we included an illustration of learning chunks in the projected motif domain to show how invariant structures can be learned by HCM.
- In response to reviewer ybpV and reviewer 2nnD, we included a justification for the choice of statistical threshold and explained and clarified how one should adapt this threshold to minimize type I or type II error.

During the discussion period, we received timely responses from reviewer SQe7 and reviewer ybpV. Unfortunately, we have not heard back from reviewer 2nnD and reviewer TVcV. We hope that reviewer TVcV and reviewer 2nnD will nonetheless find our responses and changes helpful. We certainly believe that they helped to improve our paper and appreciate their suggestions.

Thanks again to all reviewers for their time, effort, and constructive feedback.

---

### Meta-Review · Area_Chair_qEfp · 2022-08-30

**Recommendation:** Accept
**Confidence:** Certain

**Metareview:**

The paper presents a hierarchical chunking model, which builds meaningful and interpretable representations of larger units from sequential data. Effectively "parsing" the data.  The key to this model is that it more accurately predicts human response times than baselines (e.g. RNN in the original paper and PARSER in response).  The approach is also generalized to show performance on visual-spatial and fMRI domains (part-hole and sequential).  Independence testing and memory decay are the two key tunable parameters.

Primary concerns by reviewers were around clarifications (addressed) and the introduction of a stronger baseline (PARSER -- addressed).

**Award:**

No

---

### Decision · Program_Chairs · 2022-09-14

Accept